# LongMamba: Enhancing Mamba's Long Context Capabilities via Training-Free Receptive Field Enlargement

**Zhifan Ye**[1*], **Kejing Xia**[1*], **Yonggan Fu**[1,2], **Xin Dong**[2], **Jihoon Hong**[1], **Xiangchi Yuan**[1],
**Shizhe Diao**[2], **Jan Kautz**[2], **Pavlo Molchanov**[2], **Yingyan (Celine) Lin**[1,2]
[1]Georgia Institute of Technology [2]NVIDIA
{zye327,kxia39,yfu314,jhong392,xyuan300,celine.lin}@gatech.edu
{xind,sdiao,jkautz,pmolchanov}@nvidia.com

## Abstract

State space models (SSMs) have emerged as an efficient alternative to Transformer models for language modeling, offering linear computational complexity and constant memory usage as context length increases. However, despite their efficiency in handling long contexts, recent studies have shown that SSMs, such as Mamba models, generally underperform compared to Transformers in long-context understanding tasks. To address this significant shortfall and achieve both efficient and accurate long-context understanding, we propose LongMamba, a training-free technique that significantly enhances the long-context capabilities of Mamba models. LongMamba builds on our discovery that the hidden channels in Mamba can be categorized into local and global channels based on their receptive field lengths, with global channels primarily responsible for long-context capability. These global channels can become the key bottleneck as the input context lengthens. Specifically, when input lengths largely exceed the training sequence length, global channels exhibit limitations in adaptively extend their receptive fields, leading to Mamba's poor long-context performance. The key idea of LongMamba is to mitigate the hidden state memory decay in these global channels by preventing the accumulation of unimportant tokens in their memory. This is achieved by first identifying critical tokens in the global channels and then applying token filtering to accumulate only those critical tokens. Through extensive benchmarking across synthetic and real-world long-context scenarios, LongMamba sets a new standard for Mamba's long-context performance, significantly extending its operational range without requiring additional training. Our code is available at https://github.com/GATECH-EIC/LongMamba.

## 1 Introduction

The rapid advancement of large language models (LLMs) has demonstrated significant capabilities across a diverse array of real-world tasks, ranging from question answering (Zhuang et al., 2023) and document summarization (Jin et al., 2024) to code completion (Li et al., 2022). These tasks often involve processing long input sequences, such as extensive documents and sizable codebases, thereby increasing the demand for LLMs to manage increasingly longer context lengths. Contemporary commercial LLMs, including Mistral Large 2 (MistralAI, 2024) and GPT-4 (Achiam et al., 2023), feature context windows of up to 128,000 tokens.

Despite their capabilities, Transformer-based LLMs encounter significant scalability issues as sequence lengths increase (Katharopoulos et al., 2020). This is primarily due to their quadratic computational complexity and linear memory complexity as context length increases. In contrast, Mamba (Gu & Dao, 2023), one of the representative state space models (SSMs) (Gu et al., 2021; 2022a;b), offers a recurrent computation mechanism that maintains linear computational complexity and constant memory with fixed-size hidden states, enabling efficient long-context processing.

---

* Equal contribution.

However, SSMs fall short in achievable accuracy on long-context tasks compared to similarly sized Transformers, as highlighted in recent empirical studies (Waleffe et al., 2024; Ben-Kish et al., 2024).

To understand the cause of Mamba's failure to generalize to long-context lengths, we analyze the per-channel attention patterns (Ali et al., 2024) of Mamba. We find that the channels in Mamba have distinct receptive field lengths: most channels, termed *local channels*, focus on local contexts, while others, termed *global channels*, have receptive fields that extend as long as the training sequence, enabling them to capture global information from the input context. More importantly, we find that these global channels are primarily responsible for Mamba's long-context capability and can become the key bottleneck for this capability as their receptive fields fail to generalize to new sequence lengths. This failure stems from cumulative state decay that increases exponentially with context length, rendering the global channels incapable of memorizing past tokens with large decay.

Inspired by these findings and analyses, we propose LongMamba, a training-free method designed to significantly enhance the receptive fields of the identified global channels when the sequence length far exceeds the training sequence. Specifically, LongMamba enlarges the receptive fields of the identified global channels by adaptively adjusting the decay based on the target context length. This is achieved by applying token filtering to accumulate only critical tokens in the global channels' hidden state memory. This enlargement ensures that these channels can maintain their function as global information processors when exposed to much longer sequences than they were originally trained on, thereby substantially extending the functional range of Mamba models. Our contributions can be summarized as follows:

- Through visualization and analysis, we find that hidden state channels in Mamba SSMs have distinct receptive field lengths, allowing them to be categorized into local channels and global channels. We identify that the inability of global channels to capture global information when exposed to much longer sequences than they were originally trained on is the key bottleneck that limits Mamba's performance on long-context tasks.

- Building upon our findings, we propose LongMamba, a training-free method that enhances Mamba's long-context performance by effectively enlarging the receptive fields of global channels when the sequence length exceeds the training sequence. We achieve this by identifying and removing less important tokens in global channels, thereby preventing the accumulation of unimportant tokens in their hidden state memory.

- Through comprehensive benchmarking on both synthetic and real-world long-context tasks, we demonstrate that LongMamba can significantly extend the operational range of pre-trained Mamba models, outperforming previous methods aimed at enhancing Mamba's context-handling capabilities. For instance, on the widely used LongBench-E (Bai et al., 2023) dataset, our method improves task accuracy by up to $4.8\times$ compared to the vanilla Mamba models and up to $2.6\times$ over the previous approach.

## 2 RELATED WORKS

**State Space Models (SSMs).** SSMs provide a framework for representing dynamic systems through a temporal sequence of latent states, where the system's output is derived from these states (Durbin et al., 2012). In the realm of deep learning, SSMs have emerged as a promising alternative to Transformer-based architectures for sequential data processing. Initial efforts to integrate SSMs into deep learning architectures encountered significant obstacles, such as stability issues during training. The Structured State Space Sequence model (S4) (Gu et al., 2022b) marks a pivotal advancement in addressing these challenges, enabling the stable training of SSMs in deep neural networks. However, early deep SSM implementations still lack a crucial feature inherent to attention mechanisms: input-dependent information selection. Mamba (Gu & Dao, 2023) addresses this limitation by introducing selective SSM layers with input-dependent update mechanisms. A subsequent follow-up, Mamba-2 (Dao & Gu, 2024b), further refined this approach, demonstrating competitive performance as compared to Transformers. However, the difficulty of effectively handling very long-range dependencies in SSM-based models like Mamba remains a key challenge in modern language modeling, particularly when processing extended contexts beyond their initial training lengths (Ben-Kish et al., 2024; Waleffe et al., 2024).

**Mamba Models.** The Mamba architecture's efficiency and potential drive its adaptation across diverse applications. In computer vision, Vim (Zhu et al., 2024) uses bidirectional state space modeling for managing long-range dependencies in images, while VMamba (Liu et al., 2024) enhances selective SSMs with novel scanning algorithms for better information flow. DiM (Teng et al., 2024) customizes Mamba for high-resolution image diffusion. The need for extended context modeling in video, point cloud, and graph sequences boosts the demand for Mamba solutions, spurring further research. VideoMamba (Li et al., 2024) and Graph-Mamba (Wang et al., 2024a) exemplify this by applying Mamba's long temporal and spatial sequence handling. Ongoing advancements and applications further fuel the demand for Mamba's long-context capabilities. Hybrid Mamba-Attention models like Jamba (Lieber et al., 2024), Zamba(Glorioso et al., 2024b), and Hymba (Dong et al., 2024) attempt to combine the benefits of attention mechanisms with Mamba's efficiency in long-range modeling (Lieber et al., 2024; Glorioso et al., 2024b).

**Language Models for Long-Context Understanding.** Language models trained on length-limited contexts often experience performance degradation when extrapolated to longer sequences. Previous research attempted to address this problem through various approaches, including positional interpolation (Peng et al., 2024; Wang et al., 2024b), improvements to the attention mechanism (Xiao et al., 2024b; Yao et al., 2024), and external memory integration (Xiao et al., 2024a; Bulatov et al., 2022). Despite these advancements, such Transformer-based solutions frequently encounter computational and memory constraints as context lengths increase significantly. Furthermore, these methods cannot be directly applied to Mamba models due to the fundamental architectural differences between Transformers and SSMs, particularly the absence of explicit attention mechanisms in Mamba's recurrent structure. To close this gap, DeciMamba (Ben-Kish et al., 2024) is the first to explore context-extension capabilities of Mamba models. Specifically, DeciMamba employs a token pruning mechanism that progressively reduces sequence length in deeper layers by selectively removing less critical tokens, using empirically determined pruning ratios that vary across datasets and tasks. In contrast, our approach identifies the existence of global channels and their inability to capture global information when exposed to longer sequences as the key bottleneck that limits Mamba's performance on long-context tasks. Consequently, our method leverages this observation by enlarging the receptive fields of global channels, eliminating the need for meticulous layer-specific adjustments, and consistently surpassing DeciMamba in performance across diverse benchmarks.

## 3 PRELIMINARIES OF MAMBA MODELS

In this section, we provide the background of the Mamba model design and review previous efforts (Ali et al., 2024) in measuring the attention score of Mamba models, which lays the groundwork for our analysis in Sec. 4.

**Mamba model design.** Given an input sequence of $L$ tokens $I \in \mathbb{R}^{L \times d_m}$ ($d_m$ is the input channel dimension), a Mamba block maps the input sequence to output sequence $O \in \mathbb{R}^{L \times d_m}$ through the following computation:

$$X = \sigma(\text{Conv1D}(\text{Linear}_1(I))) \in \mathbb{R}^{L \times d_e} \tag{1}$$

$$Y = \text{SSM}(X) \in \mathbb{R}^{L \times d_e} \tag{2}$$

$$O = \text{Linear}_3(\sigma(\text{Linear}_2(I)) \odot Y) \in \mathbb{R}^{L \times d_m} \tag{3}$$

where $\text{Linear}_1$, $\text{Linear}_2$ and $\text{Linear}_3$ are regular linear projections, Conv1D is a 1D causal convolution with a causal mask, $\sigma$ is an activation function, and $\odot$ represents element-wise product. SSM is a state-space machine that performs a recurrent computation on the input sequence $X = (X_1, X_2, ..., X_L) \in \mathbb{R}^{L \times d_e}$ ($d_e$ is the output dimension of $\text{Linear}_1$):

$$H_t = \bar{A}_t \odot H_{t-1} + \bar{B}_t \odot X_t \in \mathbb{R}^{d_s \times d_e} \tag{4}$$

$$Y_t = C_t^T H_t \in \mathbb{R}^{d_e} \tag{5}$$

where $H_t \in \mathbb{R}^{d_s \times d_e}$ is the hidden state at time step $t$, $\bar{A}_t \in (0, 1)^{d_s \times d_e}$ is a decay factor on the hidden state, $\bar{B}_t \in \mathbb{R}^{d_s \times d_e}$ determines the hidden state update at step $t$, and $C_t \in \mathbb{R}^{d_s}$ is a per-channel output scaling factor.

The key innovation of Mamba is making $\bar{A}_t$, $\bar{B}_t$ and $C_t$ time variant (i.e., predicted from the input of the $t$-th token $X_t$). Specifically, they can be formulated as:

$$\Delta_t = \text{Softplus}(X_t), \qquad B_t, C_t = \text{Linear}_4(X_t) \in \mathbb{R}^{d_s}, \mathbb{R}^{d_s} \tag{6}$$

$$\bar{A}_t = \exp(\Delta_t \odot A), \qquad \bar{B}_t = \Delta_t \otimes B_t \tag{7}$$

where $\Delta_t \in \mathbb{R}^{d_e}_{>0}$ is a per-channel positive factor, while $A \in \mathbb{R}^{d_s \times d_e}_{<0}$ is a negative learnable matrix, which makes the hidden state decay factor $\bar{A}_t$ always smaller than 1 (i.e., continuously decaying the previous hidden state $H_{t-1}$). Finally, $\otimes$ denotes outer product operation.

**Attention Score of Mamba-based SSMs.** We can quantify the contribution of the $j$-th token's input $X_j$ to the hidden state at time step $i$ by expanding the recurrent computation in Eq. 4 across time steps:

$$H_i = \Sigma_{j=1}^{i} (\Pi_{k=j+1}^{i} \bar{A}_k) \odot \bar{B}_j \odot X_j \tag{8}$$

therefore for the output at time step $i$:

$$Y_i = C_i^T \Sigma_{j=1}^{i} (\Pi_{k=j+1}^{i} \bar{A}_k) \odot \bar{B}_j \odot X_j \tag{9}$$

$$= \Sigma_{j=1}^{i} \alpha_{i,j} \odot X_j \tag{10}$$

where we have:

$$\alpha_{i,j} = C_i^T (\Pi_{k=j+1}^{i} \bar{A}_k) \odot \bar{B}_j \in \mathbb{R}^{d_e} \tag{11}$$

is the weighting factor of the contribution of the $j$-th token's input $X_j$ to the $i$-th token's output $Y_i$, which has a similar function as the attention score in a Transformer model. Therefore, (Ali et al., 2024) proposes to regard $\alpha_{i,j}$ as the attention score between the $i$-th token and the $j$-th token. In the next section, we analyze the attention patterns of different hidden state channels (i.e., along the $d_e$ dimension). For the simplicity of notation, variables in the following sections only refers to the values at a single hidden state dimension unless otherwise noted.

# 4    ANALYZING MAMBA'S LIMITATIONS IN LONG-CONTEXT UNDERSTANDING

To understand the limited long-context understanding capabilities of Mamba models, such as their failure to retrieve passkeys (Ben-Kish et al., 2024) or look up entries in a PhoneBook (Waleffe et al., 2024) from contexts longer than the training sequence, we conduct a per-channel attention map visualization using the training sequence length in Sec. 4.1 and a longer sequence length in Sec. 4.2. This analysis lays the foundation for our proposed method in Sec. 5.

## 4.1    PER-CHANNEL RECEPTIVE FIELD CHARACTERIZATION IN VANILLA MAMBA

Fig. 1 (a) illustrates the attention map and receptive field of five sampled channels, where the red bounding boxes cover attention scores larger than $10^{-3}$, in the Mamba-130M model pre-trained on sequences of 2,000 tokens. Our empirical analysis reveals that while some channels exhibit a local attention pattern with limited receptive fields, others extend their receptive fields to cover the full sequence length. More visualizations of the per-channel receptive fields are available in Appendix A.2, which align with our observations in Fig. 1 (a).

These extensive visualizations show that there are two distinct categories of channels based on their attention patterns and receptive field patterns. Accordingly, we classify the channels within each Mamba layer into two groups: those whose receptive fields cover the training sequence length and those that do not.

**Local Channels.** These channels exhibit receptive fields that are significantly shorter than the training sequence length, as shown by channels (i), (ii), and (iii) in Fig. 1 (a). Within these channels, each token only attends to a limited local context window. Their attention pattern suggests that these channels function like a convolution layer or a sliding window attention (Beltagy et al., 2020) that captures localized information.

**Global Channels.** These channels have receptive fields that are comparable to the entire training sequence length, as shown by channels (iv) and (v) in Fig. 1 (a). In these channels, each token can effectively attend to and extract information from almost any previous token in the sequence. This means the global channels learn to capture information globally from the entire sequence.

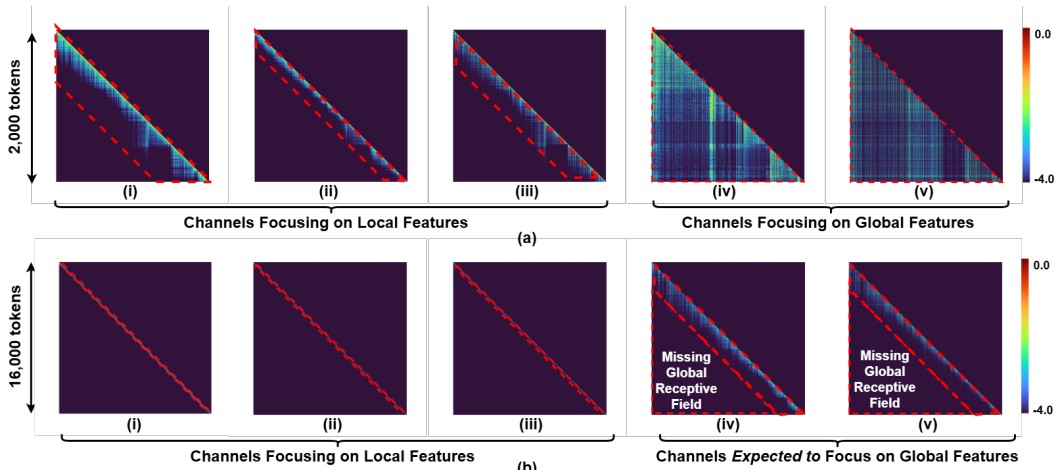

Figure 1: Visualization of the Mamba-130M model's attention map (log scale) under (a) training sequence length (2,000 tokens) and (b) extended sequence length (16,000 tokens). We uniformly sample five hidden state channels in the 12-th layer of the Mamba model and select a sequence from the Pile (Gao et al., 2020) dataset (i.e., Mamba's training dataset) as the input. The red lines delineate the receptive field of each channel, showing the range of tokens that significantly influence the current token's output. We select three channels ((i)-(iii)) with local receptive fields and two channels ((iv)-(v)) with global receptive fields to illustrate the distinct patterns of information processing in Mamba.

## 4.2 WHY MAMBA FAILS IN CONTEXT LENGTH EXTRAPOLATION?

After characterizing the distinct receptive fields of different hidden state channels, we next investigate their attention patterns at extended sequence lengths (e.g., 16,000 tokens), which are longer than the training sequence length. It can be observed from Fig. 1 (a) that although the global channels (e.g., the (iv) and (v) channels) have receptive fields covering all tokens inside the training sequence length, they fail to extend their receptive fields to 16,000 tokens (see their corresponding receptive fields in Fig. 1 (b)), rendering them unable to capture global information beyond their training sequence length. This observation is consistent with our visualization of more channels in Appendix A.2.

To dive deeper into this failure, we analyze the mechanism of hidden state updates in Mamba models. Specifically, we find that the term $\Pi_{k=j+1}^{i} \bar{A}_k$ in Eq. 8 exhibits exponential decay with growing sequence length. Take an extreme case as an example: for the hidden state $H_0$ at the initial timestep, it suffers from the following cumulative decay at the end of the sequence:

$$\Pi_{k=1}^{L} \bar{A}_k = \exp\left(\left(\sum_{k=1}^{L} \Delta_k\right) \odot A\right) \tag{12}$$

where $A$ is a negative matrix, as defined in Sec. 3. As a result, the expression above exponentially decays towards zero as the sequence length $L$ increases. Similarly, such significant decay prevents the global channels of Mamba models from preserving global information in their hidden states, when the sequence length $L$ significantly exceeds the training sequence length. The visualization of the decaying effect can be found in Appendix A.1.

## 5 THE PROPOSED LONGMAMBA FRAMEWORK

Following our analysis in Sec. 4, we propose LongMamba, a training-free technique to enhance Mamba models' long-context understanding capabilities by enlarging the receptive fields of the identified global channels. As shown in Fig. 2 (b), LongMamba features a two-step pipeline that first categorizes each hidden state channel as either a global channel or a local channel by computing the cumulative decay under the training length and then alleviates exponential hidden state decay by filtering out less important tokens from the sequence for the identified global channels. We detail these two steps in Sec. 5.1 and Sec. 5.2, respectively.

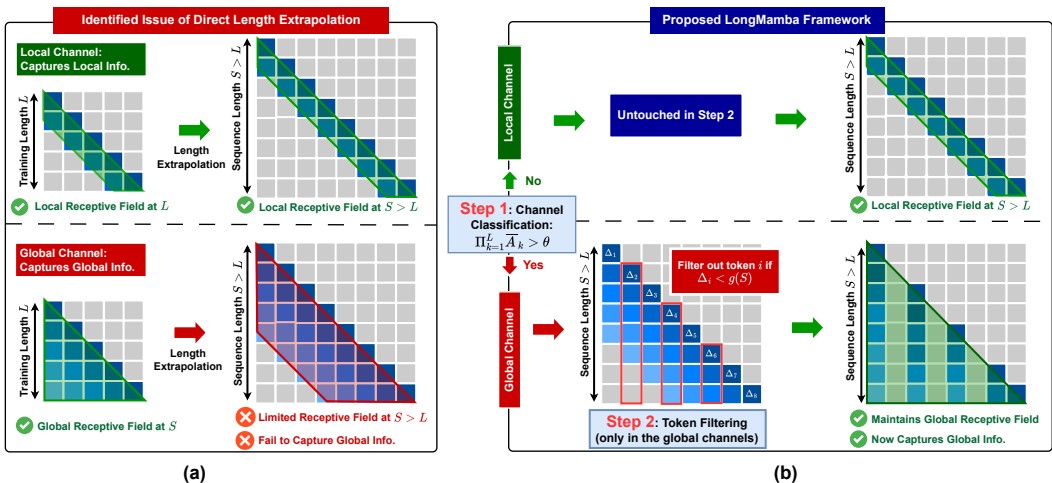

Figure 2: (a) Visualize the issue of directly applying Mamba models to a sequence (sequence length denoted as $S$) longer than the training sequence length (denoted as $L$); (b) Visualize the proposed LongMamba framework, where we enlarge the receptive fields of the global channels using the two-step pipeline detailed in Sec. 5.1 and Sec. 5.2 .

## 5.1 STEP 1: CHANNEL CLASSIFICATION

As analyzed in Sec. 4 and summarized in Fig. 2 (a), the limited receptive field of the global channels is the key bottleneck for long-context language modeling in Mamba, whereas the local channels consistently focus on local information regardless of sequence length. This motivates us to extend the receptive field only for the global channels while leaving the local channels untouched.

We distinguish between global and local channels based on the cumulative decay of each channel over the training sequence length. Specifically, if the cumulative decay factor of a channel exceeds an empirical threshold, i.e., $\Pi_{k=1}^{L}\bar{A}_k > \theta$, we classify this channel as a global channel [1], as shown in Fig. 2 (b). In other words, a channel is categorized as a global channel if it experiences hidden state decay no stronger than $\theta$ at the training length $L$.

## 5.2 STEP 2: RECEPTIVE FIELD ENLARGEMENT VIA TOKEN FILTERING

For the identified global channels, we aim to enlarge their receptive fields to ensure they can effectively capture global information from the entire input when the input sequence length $S$ is larger than the training sequence length $L$. Our key idea to achieve this is to align the cumulative decay at the input sequence length with the learned cumulative decay at the training sequence length:

$$\Pi_{i=1}^{S}\bar{A}'_i \approx \Pi_{i=1}^{L}\bar{A}_i, \tag{13}$$

where $\bar{A}'_i$ is the target decay factor we aim to achieve. This alignment can be viewed as transforming the statistics of out-of-distribution long-context inputs at test time to match those of in-distribution samples, which the global channels are capable of handling.

There are different potential ways to derive $\bar{A}'_i$ to achieve the aforementioned alignment. In this work, we adopt a simple yet effective method: token filtering for the identified global channels. Specifically, we filter out the $t$-th token for state updates when $\Delta_t$ is smaller than a certain threshold $g$, as illustrated in Fig. 2 (b). Formally, we replace $\bar{A}_t$ and $\bar{B}_t$ with $\bar{A}'_t$ and $\bar{B}'_t$ as follows:

$$(\bar{A}'_t, \bar{B}'_t) = \begin{cases} (1, 0), & \Delta_t < g, \\ (\bar{A}_t, \bar{B}_t), & \Delta_t \geq g. \end{cases} \tag{14}$$

---

[1]Because $\Pi_{k=1}^{L}\bar{A}_k \in \mathbb{R}^{d_s \times d_e}$, we average this term across the $d_s$ dimension before the comparison and classify each channel (along the $d_e$ dimension) as either a local or global channel.

When $\Delta_t < g$, we set $\bar{A}'_t = 1$ and $\bar{B}'_t = 0$, ensuring that $H_t = H_{t-1}$ (see Eq. 4). This means the hidden state is neither updated nor decayed at the $t$-th input token. For tokens with $\Delta_t \geq g$, we set $\bar{A}'_t = \bar{A}_t$ and $\bar{B}'_t = \bar{B}_t$, allowing the hidden state to be updated as usual.

For practical implementation, we must choose a suitable filtering threshold $g$ to satisfy Eq. 13. In this work, we make $g$ dependent on the sequence length $S$. Concretely, $g(S)$ is defined as a per-channel lookup table that takes $S$ as input and outputs the corresponding threshold. To construct this table, we first calibrate the distribution of $\Delta_t$ for each hidden state channel using sampled sequences from the training set (Gao et al., 2020). We then compute $g(S)$ in an offline manner such that Eq. 13 holds under the assumption that any input sequence of length $S$ follows the same distribution of $\Delta_t$.

# 6 EXPERIMENTAL RESULTS

In this section, we conduct a comprehensive evaluation of LongMamba across diverse tasks to assess its long-context understanding capabilities. Our evaluation covers three distinct datasets: language modeling (on PG-19 (Rae et al., 2019)), RULER (Hsieh et al., 2024), and LongBench-E (Bai et al., 2023). The evaluation results demonstrate LongMamba's superior performance over the previous state-of-the-art (SOTA) method (Ben-Kish et al., 2024). Additionally, we present extensive ablation studies in Sec. 6.3 to analyze the impact of hyperparameter choices and sampled calibration sequences during lookup table construction on LongMamba's performance.

## 6.1 EXPERIMENT SETTINGS

**Datasets.** We evaluate LongMamba on three datasets. To evaluate the language modeling capabilities of LongMamba, we measure its perplexity on the PG-19 (Rae et al., 2019) dataset following the settings in (Ben-Kish et al., 2024). To further assess LongMamba on more challenging tasks, we use RULER (Hsieh et al., 2024), a synthetic dataset consisting of 13 long-context tasks. In addition, we evaluate LongMamba on the tasks within LongBench-E (Bai et al., 2023), which are representative of many real-world long-context applications. For RULER, we generate 100 sequences per task and per sequence length using the official implementation.

**Models.** To evaluate LongMamba's applicability to different types of models, we apply it to two SSMs: the Mamba-1.4B (Gu & Dao, 2023) and Mamba2-1.3B (Dao & Gu, 2024a) models, as well as a hybrid Transformer-SSM model, Zamba2-1.2B (Glorioso et al., 2024a). For all experiments of this paper, we directly load the official model checkpoints without any fine-tuning or adaptation of the model weights.

**Baselines.** We compare LongMamba against two baselines: (1) the vanilla models (Gu & Dao, 2023; Dao & Gu, 2024a; Glorioso et al., 2024a) and (2) DeciMamba (Ben-Kish et al., 2024), the previous SOTA training-free method designed to enhance long-context understanding of SSMs. Since DeciMamba is implemented only for Mamba models, we report its results exclusively for the Mamba-1.4B (Gu & Dao, 2023) model. We adopt the official open-source implementations of all baselines to test their performance.

**Implementation Details.** For the hyperparameter $\theta$, which differentiates the global and local channels in Sec. 5.1, we conduct a hyperparameter search among candidate values $\{10^{-40}, 10^{-30}, 10^{-20}, 10^{-10}, 10^{-5}, 10^{-4}, 10^{-3}, 10^{-2}, 5 \times 10^{-2}, 10^{-1}, 5 \times 10^{-1}\}$ on the LongBench-E dataset and select the $\theta$ that yields the highest average accuracy. For each model, we use the same $\theta$ across all experiments. Specifically, for Mamba-1.4B, $\theta$ is set to $10^{-30}$, while for Mamba2-1.3B and Zamba2-1.2B, we set $\theta$ to $5 \times 10^{-2}$ and $10^{-5}$, respectively. When constructing the lookup table $g(S)$ used in Sec. 5.2, we first randomly sample 5 sequences from the Pile (Gao et al., 2020) dataset to calibrate the $\Delta_t$ distribution. To ensure numerical stability, we clamp extreme values of $\Delta_t$ to the top $C\%$ largest values. We search for the optimal $C$ among candidates $\{0, 5, 10, 15, 20\}$ following the same procedure as for $\theta$. As a result, $C$ is set to 5 for both Mamba2-1.3B and Zamba2-1.2B and 20 for Mamba-1.4B. We then precompute the lookup entries at 1000-token intervals (i.e., 1000, 2000, 3000, etc.). Before looking up the table, we always round the sequence length $S$ to the nearest 1000-token interval.

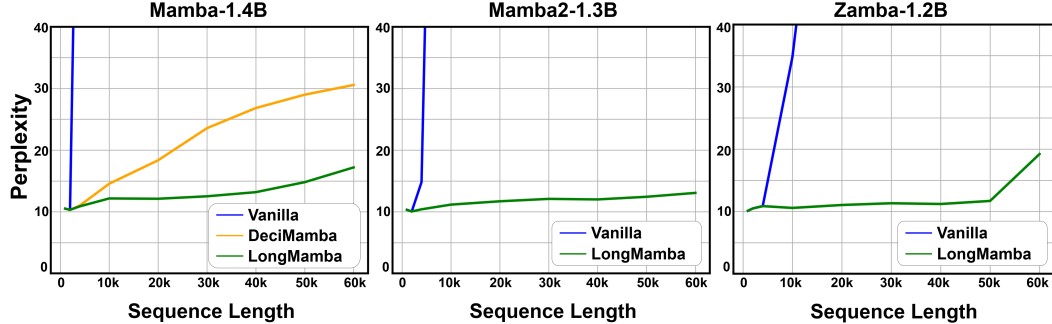

Figure 3: Perplexity on the PG-19 dataset under varying sequence lengths. We evaluate three models: Mamba-1.4B, Mamba2-1.3B, and Zamba2-1.2B. The Mamba-1.4B and Mamba2-1.3B models are trained on sequences of 2k tokens, while the Zamba2-1.2B model is trained on sequences of 4k tokens. For all three models, we measure perplexity on PG-19 sequences of up to 60k tokens. In the figure, "Vanilla" refers to the baseline models without applying DeciMamba or LongMamba.

Table 1: Task accuracy (%) on the RULER dataset using the vanilla Zamba2-1.2B model (referred to as "Vanilla" in the table) and the Zamba2-1.2B model enhanced with LongMamba across different sequence lengths (16k, 24k, and 32k tokens). For task name abbreviations, please refer to Appendix D for details. Here we omit the MK2 and MK3 tasks for both the baseline and Long-Mamba, as both methods struggle to achieve meaningful (i.e., nonzero) accuracy.

| Length | Method | S1 | S2 | S3 | MK1 | MV | MQ | VT | CWE | FWE | QA1 | QA2 | Avg. |
|--------|--------|------|------|------|------|------|------|------|------|------|------|------|------|
| 16k | Vanilla | 30.00 | 11.00 | 7.00 | 6.00 | 8.00 | 0.00 | 0.20 | 0.10 | 13.67 | 0.00 | 1.00 | 7.00 |
| | LongMamba | 79.00 | 92.00 | 31.00 | 23.00 | 58.00 | 49.25 | 0.20 | 2.30 | 0.67 | 1.00 | 11.00 | **31.58** |
| 24k | Vanilla | 43.00 | 9.00 | 0.00 | 8.00 | 7.50 | 0.25 | 0.00 | 0.00 | 1.67 | 1.00 | 7.00 | 7.04 |
| | LongMamba | 56.00 | 79.00 | 29.00 | 25.00 | 20.50 | 18.00 | 7.00 | 0.30 | 1.67 | 2.00 | 6.00 | **22.22** |
| 32k | Vanilla | 26.00 | 0.00 | 0.00 | 0.00 | 0.00 | 0.00 | 1.80 | 0.10 | 1.00 | 0.00 | 1.00 | 2.72 |
| | LongMamba | 34.00 | 73.00 | 16.00 | 17.00 | 0.80 | 0.50 | 4.00 | 0.20 | 0.67 | 2.00 | 4.00 | **13.83** |

## 6.2 EVALUATING LONGMAMBA ON LONG-CONTEXT TASKS

**Language Modeling.** First, we test the perplexity of LongMamba and the baseline methods on the PG-19 dataset (Rae et al., 2019), following the same experiment setting as DeciMamba (Ben-Kish et al., 2024). We use three pre-trained models—Mamba-1.4B and Mamba2-1.3B (both trained on sequences of 2k tokens) and Zamba2-1.2B (trained on sequences of 4k tokens)—and apply them to sequences of up to 60k tokens. As shown in Fig. 3, LongMamba consistently outperforms both the vanilla models and the previous SOTA method, DeciMamba, across all tested sequence lengths. Specifically, the perplexity of the vanilla models rapidly increases to more than 40 as the sequence length exceeds 10k tokens. While DeciMamba slows down the growth of perplexity compared to vanilla models, it still exceeds 30 at a sequence length of 60k tokens when applied to the Mamba-1.4B model. In contrast, across all three tested models, LongMamba consistently keeps the perplexity below 20, demonstrating its effectiveness in long-context modeling. We hypothesize that LongMamba's superior performance, compared to DeciMamba—which prunes tokens across all hidden state channels—stems from its differentiated treatment of global and local channels.

**RULER.** To better understand LongMamba's performance on specific long-context tasks such as passkey retrieval, we further benchmark it on the RULER dataset (Hsieh et al., 2024), which comprises multiple unique long-context tasks, including passkey retrieval and question answering. Tab. 1 details the performance of LongMamba applied to the Zamba2-1.2B model. We observe that at sequence lengths of 16k, 24k, and 32k, the proposed method achieves improvements of 24.58%, 15.18% and 11.11% in average accuracy, respectively, demonstrating LongMamba's effectiveness across varying sequence lengths. The most notable performance improvement is on the S2 task, which requires the model to retrieve passkey tokens from a long, irrelevant context sampled from Paul Graham's essays (Kamradt, 2023). Here, LongMamba improves task accuracy from 0% to 73% for sequences of 32k tokens, demonstrating the strong retrieval capabilities of LongMamba.

Table 2: Task accuracy (%) on the LongBench-E dataset when applying LongMamba and the baselines to different models. Here, "Vanilla" refers to the baseline models without DeciMamba or LongMamba. For task name abbreviations, refer to Appendix D for details. We categorize tasks into different types (e.g., Single-doc QA) following the classification used in LongBench-E.

| Model | Method | Synthetic | | Summary | | Single-doc QA | | Multi-doc QA | | Few-shot Learning | | | Coding | | Avg. |
|---|---|---|---|---|---|---|---|---|---|---|---|---|---|---|---|
| | | PC | PR | GR | MN | MQA | QA | 2WM | HQA | SS | TR | TQA | LCC | RB | |
| **Mamba-1.4B** | Vanilla | 1.41 | 4.59 | 7.41 | 8.19 | 7.97 | 3.91 | 7.40 | 4.43 | 4.72 | 15.67 | 17.29 | 16.39 | 9.45 | 8.37 |
| | DeciMamba | 0.80 | 5.44 | 9.17 | 10.39 | 8.82 | 4.01 | 6.99 | 5.39 | 8.60 | 14.33 | 29.70 | 39.02 | 31.32 | 13.38 |
| | LongMamba | 1.00 | 4.77 | 11.74 | 8.93 | 10.60 | 2.92 | 8.73 | 6.41 | 7.00 | 37.00 | 40.63 | 46.29 | 39.26 | **17.33** |
| **Mamba2-1.3B** | Vanilla | 1.29 | 0.81 | 7.67 | 5.84 | 11.45 | 2.19 | 2.31 | 2.88 | 4.69 | 14.67 | 10.08 | 22.94 | 19.89 | 8.21 |
| | LongMamba | 2.02 | 3.51 | 14.33 | 10.28 | 14.73 | 5.14 | 5.73 | 5.52 | 14.00 | 21.67 | 48.74 | 42.99 | 36.75 | **17.34** |
| **Zamba2-1.2B** | Vanilla | 5.72 | 1.75 | 9.31 | 5.39 | 4.30 | 4.53 | 4.42 | 6.29 | 9.80 | 33.33 | 28.14 | 15.56 | 20.07 | 11.43 |
| | LongMamba | 3.33 | 3.67 | 10.42 | 8.67 | 8.92 | 5.85 | 6.64 | 8.29 | 25.79 | 39.00 | 63.02 | 22.39 | 25.63 | **17.82** |

Table 3: Task accuracy (%) of LongMamba on the Longbench-E dataset with different channel selection thresholds ($\theta$) when applied to the Mamba2-1.3B model. For task name abbreviations, refer to Appendix D for details.

| $\theta$ | PC | PR | GR | MN | MQA | QA | 2WM | HQA | SS | TR | TQA | LCC | RB | Avg. |
|---|---|---|---|---|---|---|---|---|---|---|---|---|---|---|
| $7.5 \times 10^{-2}$ | 0.71 | 3.99 | 13.05 | 10.07 | 15.15 | 3.63 | 4.81 | 4.85 | 16.85 | 30.33 | 44.84 | 38.97 | 31.29 | 16.81 |
| $5.0 \times 10^{-2}$ | 2.02 | 3.51 | 14.33 | 10.28 | 14.73 | 5.14 | 5.73 | 5.52 | 14.00 | 21.67 | 48.74 | 42.99 | 36.75 | 17.34 |
| $2.5 \times 10^{-2}$ | 1.92 | 6.55 | 14.41 | 10.50 | 14.45 | 5.02 | 5.42 | 4.84 | 18.67 | 44.56 | 40.58 | 32.38 | 16.14 |
| $1.0 \times 10^{-2}$ | 2.95 | 6.87 | 14.93 | 10.71 | 13.84 | 4.62 | 5.02 | 4.83 | 11.17 | 22.67 | 44.19 | 41.03 | 33.08 | 16.61 |
| $7.5 \times 10^{-3}$ | 2.94 | 6.47 | 14.50 | 10.54 | 14.01 | 4.26 | 5.25 | 4.80 | 10.19 | 23.00 | 44.04 | 39.77 | 32.26 | 16.31 |
| $5.0 \times 10^{-3}$ | 2.77 | 6.29 | 14.49 | 10.36 | 14.05 | 4.61 | 5.16 | 4.90 | 10.11 | 24.33 | 45.41 | 39.61 | 32.02 | 16.47 |
| $2.5 \times 10^{-3}$ | 2.55 | 6.37 | 14.16 | 10.11 | 13.74 | 4.50 | 4.68 | 4.83 | 11.08 | 20.00 | 45.44 | 40.30 | 32.59 | 16.18 |
| $1.0 \times 10^{-3}$ | 2.80 | 6.05 | 14.24 | 9.97 | 13.62 | 4.43 | 4.76 | 4.70 | 13.02 | 18.00 | 41.15 | 40.77 | 33.57 | 15.93 |
| $7.5 \times 10^{-4}$ | 2.79 | 5.87 | 13.79 | 9.81 | 13.77 | 4.63 | 4.26 | 4.75 | 12.32 | 19.67 | 42.48 | 41.18 | 33.38 | 16.05 |
| $5.0 \times 10^{-4}$ | 2.99 | 5.71 | 13.94 | 10.26 | 13.23 | 4.41 | 4.16 | 4.45 | 11.44 | 17.33 | 43.68 | 40.47 | 32.99 | 15.77 |
| $2.5 \times 10^{-4}$ | 2.44 | 5.42 | 14.19 | 9.94 | 12.93 | 4.64 | 4.76 | 4.43 | 12.12 | 17.00 | 42.77 | 42.49 | 34.12 | 15.94 |
| $1.0 \times 10^{-4}$ | 1.95 | 5.22 | 14.59 | 10.58 | 12.90 | 4.73 | 4.72 | 4.37 | 15.58 | 13.33 | 42.08 | 40.64 | 33.54 | 15.71 |
| $7.5 \times 10^{-5}$ | 2.00 | 5.51 | 14.50 | 10.32 | 13.05 | 4.76 | 4.48 | 4.54 | 15.40 | 13.67 | 42.40 | 40.95 | 33.48 | 15.77 |
| $5.0 \times 10^{-5}$ | 1.98 | 5.44 | 14.51 | 10.17 | 12.77 | 4.77 | 4.66 | 4.49 | 15.37 | 13.00 | 41.72 | 40.42 | 33.97 | 15.64 |
| $2.5 \times 10^{-5}$ | 3.11 | 6.10 | 14.47 | 9.42 | 13.10 | 4.72 | 4.62 | 4.48 | 14.82 | 13.00 | 42.07 | 40.74 | 32.78 | 15.65 |
| $1.0 \times 10^{-5}$ | 3.10 | 5.53 | 14.53 | 9.78 | 12.85 | 4.90 | 4.33 | 4.69 | 12.17 | 12.00 | 38.10 | 39.37 | 31.91 | 14.87 |

**LongBench-E.** In addition to the language modeling task and synthetic tasks from RULER, we further test LongMamba on the LongBench-E dataset (Bai et al., 2023), which covers 13 long-context applications, such as coding and few-shot learning. The benchmark results for all tasks are presented in Tab. 2, where we evaluate the performance of LongMamba and the baselines on the Mamba-1.4B, Mamba2-1.3B and Zamba2-1.2B models. The results show that LongMamba improves the average accuracy by 8.76%, 9.13% and 6.39% compared to the vanilla Mamba-1.4B, Mamba2-1.3B and Zamba2-1.2B models, respectively. Compared to the previous SOTA DeciMamba, LongMamba increases the average accuracy from 13.38% to 17.33% on the Mamba-1.4B model. One of the most notable performance improvements is observed on the LCC task, where the proposed Long-Mamba enhances the task accuracy from 16.39% with the vanilla Mamba-1.4B model to 46.29%, highlighting its potential for real-world applications such as coding. Furthermore, by comparing the long-context performance of pure SSMs (Mamba-1.4B and Mamba2-1.2B) and the hybrid model (Zamba2-1.2B), we find that: (1) although the vanilla hybrid model has more than 3% higher average accuracy than the pure SSMs, LongMamba closes the gap between the two types of models; and (2) the two types of models excel at different types of long-context tasks. For example, the Mamba-1.4B and Mamba2-1.3B models perform better on the coding task LCC while the Zamba2-1.2B model achieves the highest accuracy on the few-shot learning task TQA. Notably, the proposed LongMamba enhances the performance of both LCC and TQA for both types of models, further validating its effectiveness.

## 6.3 ABLATION STUDIES OF LONGMAMBA

In this subsection, we conduct two ablation studies on LongMamba: (1) the performance impact of adopting different channel selection thresholds $\theta$ in Sec. 5.1; and (2) the robustness of LongMamba against different randomly sampled sequences during the calibration process described in Sec. 5.2.

**Ablation Study on Channel Selection Thresholds.** To understand the impact of different threshold values $\theta$, we conduct an ablation study on the LongBench-E dataset (Bai et al., 2023) using the

Table 4: Task accuracy (%) of the proposed LongMamba on the LongBench-E dataset with different calibration sequence groups sampled from the Pile dataset. Here LongMamba is applied to the Mamba2-1.3B model. Indexes 0 to 9 in the table refer to different sequence groups; MEAN and STD represent the mean and standard deviation of task accuracy across different groups, respectively. For task name abbreviations, refer to Appendix D for details. Index 0 corresponds to the sequence group used for all other benchmarks in the paper.

| Index | PC | PR | GR | MN | MQA | QA | 2WM | HQA | SS | TR | TQA | LCC | RB | Avg. |
|-------|------|------|-------|-------|-------|------|------|------|-------|-------|-------|-------|-------|-------|
| 0 | 2.02 | 3.51 | 14.33 | 10.28 | 14.73 | 5.14 | 5.73 | 5.52 | 14.00 | 21.67 | 48.74 | 42.99 | 36.75 | 17.34 |
| 1 | 1.67 | 5.48 | 14.78 | 10.97 | 14.70 | 4.65 | 5.10 | 5.48 | 13.52 | 22.00 | 47.21 | 42.42 | 36.05 | 17.23 |
| 2 | 0.93 | 3.97 | 14.02 | 10.88 | 15.00 | 4.96 | 5.93 | 5.46 | 18.99 | 27.00 | 49.47 | 43.46 | 37.55 | 18.28 |
| 3 | 1.67 | 5.03 | 14.30 | 10.52 | 14.53 | 4.66 | 5.51 | 5.28 | 10.55 | 21.67 | 49.75 | 42.87 | 36.79 | 17.16 |
| 4 | 1.05 | 3.92 | 14.23 | 10.63 | 14.59 | 5.00 | 4.77 | 5.53 | 13.23 | 21.67 | 46.12 | 44.98 | 35.66 | 17.03 |
| 5 | 1.47 | 3.86 | 14.44 | 11.06 | 14.47 | 5.12 | 5.51 | 5.51 | 17.46 | 28.67 | 45.90 | 44.75 | 36.38 | 18.05 |
| 6 | 1.41 | 2.88 | 13.15 | 10.76 | 15.91 | 4.63 | 4.51 | 5.17 | 17.06 | 30.33 | 46.73 | 43.21 | 34.98 | 17.75 |
| 7 | 2.14 | 3.27 | 14.10 | 9.46 | 14.77 | 4.91 | 5.16 | 5.46 | 12.22 | 27.67 | 45.55 | 41.95 | 34.20 | 16.99 |
| 8 | 1.84 | 6.12 | 14.32 | 10.60 | 15.10 | 4.81 | 4.81 | 4.97 | 10.96 | 21.00 | 47.69 | 43.61 | 36.24 | 17.08 |
| 9 | 1.35 | 3.05 | 14.26 | 11.09 | 15.26 | 4.96 | 5.34 | 5.48 | 15.08 | 22.00 | 47.32 | 44.09 | 37.15 | 17.42 |
| MEAN | 1.56 | 4.11 | 14.19 | 10.63 | 14.91 | 4.88 | 5.24 | 5.39 | 14.31 | 24.37 | 47.45 | 43.43 | 36.18 | 17.43 |
| STD | 0.37 | 1.03 | 0.40 | 0.46 | 0.41 | 0.18 | 0.43 | 0.18 | 2.68 | 3.41 | 1.40 | 0.92 | 0.96 | 0.42 |

Mamba2-1.3B model across a wide range of threshold choices, ranging from $7.5 \times 10^{-2}$ to $1 \times 10^{-5}$. As shown in Tab. 3, LongMamba is robust to different threshold choices; for instance, the accuracy gap remains within 1.2% for any tested $\theta$ values between $7.5 \times 10^{-2}$ and $7.5 \times 10^{-3}$. Furthermore, among all tested threshold values, we find that $5 \times 10^{-2}$ consistently achieves the best performance on the Longbench-E dataset. Therefore, we adopt $\theta = 5 \times 10^{-2}$ as the threshold setting for all datasets when applying LongMamba to the Mamba2-1.2B model.

**Ablation Study on Calibration Sequences.** To assess the robustness of LongMamba's calibration process in Sec. 5.2, we evaluate its performance on the LongBench-E dataset (Bai et al., 2023) when applied to the Mamba2-1.3B model with different groups of sampled sequences. Specifically, in this ablation study, we randomly sample 10 groups of sequences from the Pile dataset (Gao et al., 2020), each consisting of 5 sampled sequences at the training sequence length of the Mamba2-1.3B model (i.e., 2k tokens). As shown in Tab. 4, LongMamba demonstrates stable performance across different calibration sequence groups. The standard deviation (STD) of the average accuracy on the LongBench-E dataset remains within 0.42%. Furthermore, across all tasks in the LongBench-E dataset, the STD of task accuracy is consistently within 3.41%. These results demonstrate the robustness of the proposed LongMamba with respect to the choice of calibration samples.

## 7 CONCLUSION

In this paper, we present LongMamba, a training-free method designed to enhance the capabilities of Mamba SSMs for long-context tasks. Our approach builds on several key findings: First, we discover that hidden state channels in Mamba models exhibit distinct receptive field lengths—local channels focus on nearby contexts, while global channels engage with the entire input sequence. Second, we identify that the inability of global channels to handle global information from sequences longer than their training length—due to exponential hidden state decay—is the key bottleneck limiting Mamba's effectiveness in long-context tasks. Finally, our analysis demonstrates that analytically identifying and removing less important tokens in global channels can adaptively alleviate the exponential decay of hidden states, which intensifies with increased context length, thereby significantly expanding these channels' receptive fields. Through extensive benchmarking across synthetic and real-world long-context scenarios, LongMamba sets a new standard for Mamba in long-context tasks. Our findings enhance the understanding of Mamba and may inspire new innovations for long-context models.

## ACKNOWLEDGMENT

This work was partially supported by the National Science Foundation (NSF) Computing and Communication Foundations (CCF) program (Award ID: 2400511), and CoCoSys, one of the seven centers in JUMP 2.0, a Semiconductor Research Corporation (SRC) program sponsored by DARPA.

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

# A  MORE VISUALIZATIONS

## A.1  VISUALIZING THE HIDDEN STATE DECAY

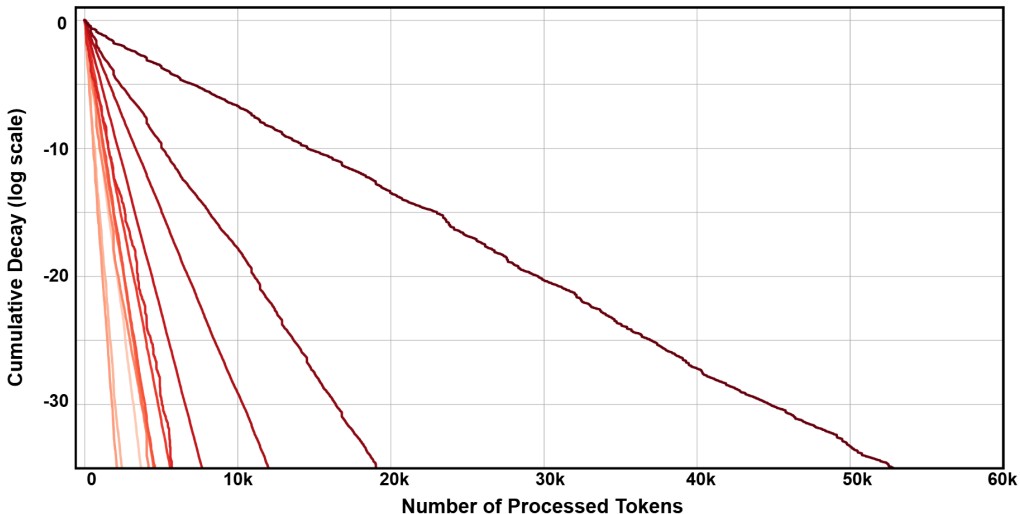

Figure 4: Visualization of the cumulative hidden state decay (defined in Eq. 12) as the number of to-kens processed by the model increases. In the figure, we visualize 12 global channels sampled from the 16th layer of the Mamba-130M model. We use a sequence sampled from the Pile dataset(Gao et al., 2020) as input and plot the hidden state decay of each global channel in a unique color.

## A.2 Visualizing more Attention Maps

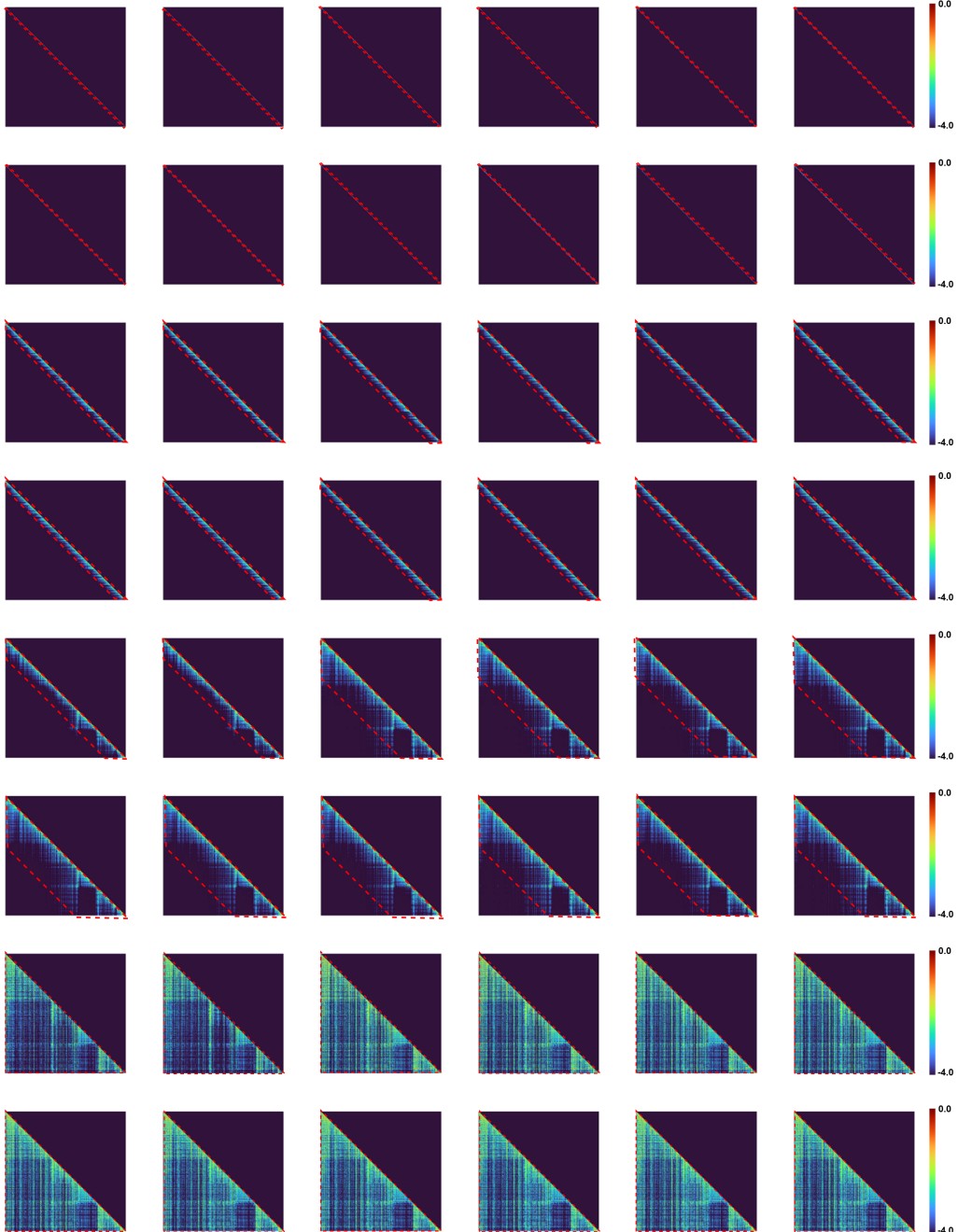

Figure 5: Visualization of the attention maps (log scale) for a sequence composed of 2,000 tokens. In this figure, we sample a sequence from the Pile (Gao et al., 2020) dataset as input and visualize 48 channels randomly sampled from the Mamba-130M model. The channels are sorted by their cumulative decay on the sampled sequence. The red lines delineate the receptive field of each channel, covering all attention scores greater than $10^{-3}$.

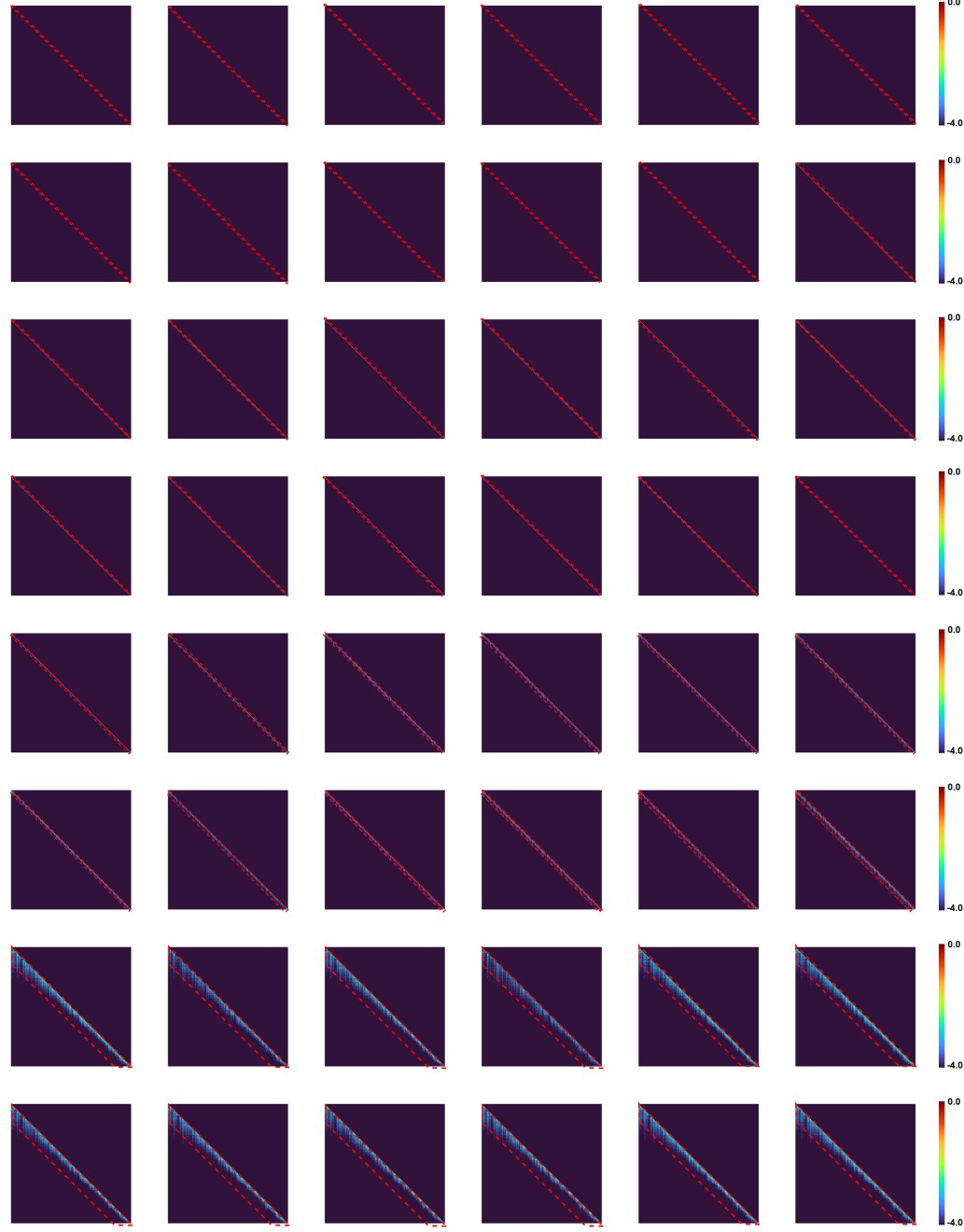

Figure 6: Visualization of the attention maps (log scale) for a sequence composed of 16,000 tokens. In this figure, we use the same input sequence and channels as in the previous Fig. 5. The red lines delineate the receptive field of each channel, covering all attention scores greater than $10^{-3}$. We observe that the last two rows of channels, although they have a receptive field covering all 2,000 tokens in Fig. 5, struggle to extend their receptive field to the entire sequence of 16,000 tokens in this figure.

Table 5: Task accuracy of the vanilla Falcon-Mamba-7B model (Zuo et al., 2024), the LongMamba-enhanced Falcon-Mamba-7B model, and the Transformer baselines (i.e., Llama2-7B-chat-4k (Touvron et al., 2023), XGen-7B-8k (Nijkamp et al., 2023), and Vicuna-v1.5-7B-16k (Zheng et al., 2023)) on LongBench. To enhance readability, the table is split into two parts: the top half and the bottom half. Here, "Vanilla" refers to the vanilla Falcon-Mamba-7B model without LongMamba. For task name abbreviations, refer to Appendix D for details.

| Datasets | PC | SS | 2WM | TQA | QA | VCS | MSQ | MN | QMS | HQA | LCC |
|---|---|---|---|---|---|---|---|---|---|---|---|
| Vanilla | 0.1 | 20.0 | 27.6 | 73.1 | 31.0 | 7.2 | 7.4 | 25.8 | 18.5 | 22.6 | 48.4 |
| LongMamba | 1.5 | 25.1 | 28.2 | 79.9 | 32.1 | 9.9 | 12.1 | 25.8 | 21.5 | 31.8 | 49.5 |
| Llama2-7B-chat-4k | 2.1 | 40.7 | 32.8 | 77.8 | 19.2 | 0.2 | 9.4 | 25.8 | 20.8 | 25.4 | 52.4 |
| XGen-7B-8k | 2.1 | 25.3 | 21.1 | 77.8 | 18.1 | 2.2 | 10.3 | 26.2 | 20.5 | 29.7 | 38.6 |
| Vicuna-v1.5-7B-16k | 6.5 | 40.8 | 20.8 | 86.2 | 26.1 | 15.1 | 9.8 | 27.2 | 22.8 | 25.3 | 51.0 |
| Datasets | NQA | DR | MQAzh | MQA | GR | LS | RB | PR | PRzh | TR | Avg. |
| Vanilla | 5.9 | 12.1 | 20.0 | 30.2 | 22.5 | 2.3 | 38.6 | 4.5 | 3.3 | 70.0 | 23.4 |
| LongMamba | 13.2 | 11.3 | 20.2 | 30.8 | 23.9 | 12.3 | 42.4 | 4.5 | 3.3 | 70.0 | 26.2 |
| Llama2-7B-chat-4k | 18.7 | 5.2 | 18.9 | 36.8 | 27.3 | 19.8 | 43.8 | 9.8 | 0.5 | 61.5 | 26.1 |
| XGen-7B-8k | 18.0 | 11.0 | 14.8 | 37.7 | 27.3 | 20.5 | 38.6 | 8.5 | 3.5 | 65.5 | 24.6 |
| Vicuna-v1.5-7B-16k | 19.4 | 19.3 | 43.0 | 38.5 | 27.9 | 28.8 | 43.5 | 4.5 | 5.0 | 71.5 | 30.1 |

Table 6: Comparison of the prefilling latency on A5000 between the vanilla models (Gu & Dao, 2023; Dao & Gu, 2024a; Glorioso et al., 2024a) and the corresponding LongMamba-enhanced models across various sequence lengths (4k, 8k, 16k, 24k, 32k, and 40k tokens). The batch size for all experiments is set to 1. The unit of all latency measurements is seconds.

| Model | Method | 4k | 8k | 16k | 24k | 32k | 40k | Avg. |
|---|---|---|---|---|---|---|---|---|
| **Mamba-1.4B** | Vanilla | 0.8424 | 1.7395 | 3.2559 | 4.9148 | 6.5405 | 8.1963 | 4.2482 |
| | LongMamba | 0.9605 | 1.9038 | 3.4392 | 5.1085 | 6.7618 | 8.4568 | 4.4384 |
| **Mamba2-1.3B** | Vanilla | 0.8533 | 1.6480 | 3.4492 | 4.8757 | 6.8910 | 8.6408 | 4.3930 |
| | LongMamba | 0.9445 | 1.7594 | 3.5713 | 5.0021 | 6.9652 | 8.7078 | 4.4917 |
| **Zamba2-1.2B** | Vanilla | 0.3365 | 0.7493 | 1.8686 | 3.3916 | 5.2869 | 7.6480 | 3.2135 |
| | LongMamba | 0.3445 | 0.8443 | 1.9448 | 3.4730 | 5.3659 | 7.7433 | 3.2860 |

# B MORE BENCHMARK RESULTS

In Tab. 5, we apply LongMamba to an 8B SSM, Falcon-Mamba-7B, and compare its performance with the vanilla Falcon-Mamba-7B model and several Transformer baselines of similar sizes. The results yield the following observations: (1) The vanilla Mamba exhibits lower performance compared to Transformer models on long context understanding tasks. Specifically, Llama2-7B-chat-4k (Touvron et al., 2023), XGen-7B-8k (Nijkamp et al., 2023), and Vicuna-v1.5-7B-16k (Zheng et al., 2023) achieve 2.7%, 1.3%, and 6.7% higher average accuracy, respectively; (2) LongMamba improves the vanilla Mamba model's performance, narrowing the the gap with or even surpassing some of the Transformer models. As shown in the last column of the bottom half of the table, LongMamba increases average accuracy by 2.8% over the vanilla Mamba model, outperforming Llama2-7B-chat-4k by 0.1% and reducing the gap with Vicuna-v1.5-7B-16k from 6.7% to 4.0%. These results suggest a promising direction for the research community, indicating that Mamba's long-context capabilities can rival those of Transformers.

# C LATENCY OVERHEAD OF LONGMAMBA

This section evaluates the latency overhead of LongMamba during inference when applied to three models: Mamba-1.4B, Mamba2-1.3B, and Zamba-1.2B. Tab. 6 compares the prefilling latency of the vanilla and LongMamba-enhanced models across different sequence lengths. The results indicate that LongMamba introduces a small latency overhead, with an average increase in prefilling latency of at most 4.5%.

## D  TASK NAME ABBREVIATIONS

**LongBench-E Tasks:**

- 2WM: 2WikiMQA
- GR: GovReport
- HQA: HotpotQA
- LCC: LCC
- MQA: MultiFieldQA-en
- MN: MultiNews
- PC: Passage Count
- PR: PassageRetrieval-en
- QA: Qasper
- RB: RepoBench-P
- SS: SAMSum
- TR: TREC
- TQA: TriviaQA

**LongBench Tasks:**

- PC: Passage Count
- SS: SAMSum
- 2WM: 2WikiMQA
- TQA: TriviaQA
- QA: Qasper
- VCS: VCSUM (zh)
- MSQ: Musique
- MN: MultiNews
- QMS: QMSum
- HQA: HotpotQA
- LCC: LCC
- NQA: NarrativeQA
- DR: DuReader (zh)
- MQAzh: MultiFieldQA-zh
- MQA: MultiFieldQA-en
- GR: GovReport
- LS: LSHT (zh)
- RB: RepoBench-P
- PR: PassageRetrieval-en
- PRzh: PassageRetrieval-zh
- TR: TREC

**RULER Tasks:**

- S1-S3: single1-3
- MK1-MK3: multikey1-3
- MV: multivalue

- MQ: multiquery
- VT: vt
- CWE: cwe
- FWE: fwe
- QA1-QA2: qa1-2

