# OpenReview forum: "LongMamba: Enhancing Mamba's Long-Context Capabilities via Training-Free Receptive Field Enlargement"
_ICLR.cc/2025/Conference — ICLR 2025 Poster_

### Official Review · Reviewer_1AHb · 2024-11-02

**Soundness:** 2
**Presentation:** 3
**Contribution:** 2
**Rating:** 6
**Confidence:** 3

**Summary:**

This paper introduces LongMamba, which is a training-free method to enhance Mamba long context performance. It identified that the channels in Mamba have distinct receptive field lengths: local and global. And it hypothesize that this failure of the global channels
to capture global information under extended sequence lengths is the root cause of Mamba’s poor performance in long-context understanding. LongMamba expands the global channels' receptive fields by removing less important tokens.

**Strengths:**

1. The discovery and explanation of  Mamba models’ limited performance in long contexts are quite interesting.

2. The paper conducts extensive experiments, and LongMamba show its effectiveness

**Weaknesses:**

1. While I really appreciate the discovery and root analysis of Mamba's limitations regarding long contexts, I find that LongMamba's approach—pruning less important tokens to increase the global channel's receptive field—seems somewhat incremental compared to DeciMamba.

2. LongMamba shows some improvements for long contexts, but it still seems to be somewhat behind transformer-based LLMs designed for long contexts. I understand that the authors are currently using a training-free approach. If training were allowed, would the performance be further enhanced?

3. It would be more convincing if the authors can provide an evaluation on the Ruler benchmark[1].

[1] https://github.com/NVIDIA/RULER

**Questions:**

see the weakness section

---

> ### Author Response · Authors · 2024-11-25
> **Author Response #1**
>
> We greatly appreciate your positive feedback and constructive suggestions/questions, and have addressed each of your concerns and suggestions below:
>
> ## **Q1: I find that LongMamba's approach—pruning less important tokens to increase the global channel's receptive field—seems somewhat incremental compared to DeciMamba.**
>
> We humbly clarify that the key contribution of LongMamba lies in the insight distinguishing **global** and **local** channels in Mamba, which serves as the foundation for enhancing its long-context capabilities. While we use token filtering to implement this insight, we emphasize that it is just one of many possible approaches to leverage or implement our insight, showcasing the broader applicability and originality of our contribution. Specifically, our new contributions beyond DeciMamba are as follows:
>
> **1. Key Insight:**
>
> We discover that the exponential hidden state decay of the global channels is the primary cause of Mamba’s constrained receptive field and limited performance on long-context tasks.
> This finding is also highly recognized by other reviewers: Reviewer P417 remarked, "The finding about the categroies of channels of Mamba models (local and global) is interesting and meaningful, which could bring insight to the community for better understanding the Mamba model's behavior", and Reviewer GwXu stated, “the main result is very interesting.”
>
> **2. Token Filtering Implementation:**
>
> Leveraging this key insight, we propose specially handling the global channels to enlarge their receptive field by applying selective token filtering exclusively to the global channels while preserving the locality of the local channels. This strategy has outperformed DeciMamba and maintained the same language modeling perplexity on contexts 10x longer, further validating the **significance and practicality** of our key insight. This has also been recognized by other reviews, such as “Overall a simple mechanism that seemingly enables Mamba to generalize to much longer contexts” by Reviewer GwXu.
>
> **3. Alternative Implementation:**
>
> To further demonstrate the generality of our insight, during the rebuttal period, we implemented and validated an alternative method.
>
> This method dubbed “decay normalization” adjusts the decay by normalizing $\Delta_t$ for all tokens in the same global channel with a scalar constant c (i.e., $\Delta’_t = \Delta_t / c$), and subsequently derives the adjusted $\bar{A}’_t$ and $\bar{B}’_t$ using Eq. 7 of the submitted manuscript. The factor $c=c(S)$ is computed to satisfy Eq. 16 in Sec. 5.2 of the submitted manuscript, similar to how we obtain $g(S)$.
>
> We evaluated this alternative decay adjustment method on both the LongBench and PG19 tasks. The results are summarized in Tab. A and Tab. B below, where “Ours: Token Filtering” corresponds to the decay adjustment method described in the submitted manuscript, and “Ours: Normalization” represents the alternative approach proposed above.

---

> ### Author Response · Authors · 2024-11-25
> **Author Response #2**
>
> **Tab. A: Benchmarking results on LongBench dataset with 1.4B Mamba models.** This table compares the LongBench performance achieved by the vanilla Mamba model, DeciMamba, as well as the two proposed methods that specially handle the global channels stemming from our insight. To enhance readability, the table is divided into two sections (top and bottom), with all values expressed as percentages.
>
> | Datasets | Passage Count | SAMSum | 2WikiMQA | TriviaQA | Qasper | VCSUM (zh) | Musique | MultiNews | QMSum | HotpotQA | LCC |
> |----|--|--|----|----|---|----|---|---|---|---|---|
> | Metrics | Accuracy | Rouge-L | F1 | F1 | F1 | Rouge-L | F1 | Rouge-L | Rouge-L| F1 | Edit Sim|
> | Vanilla Mamba | 0.4 | 3.7 | 8.6 | 11.4 | 13.6 | 2.1 | 0.4 | 15.7 | 1.1 | 4.7 | 40.6 |
> | DeciMamba | 0.1 | 8.7 | 6.5 | 23.4 | 14.2 | 3.2 | 1.4 | 15.8 | 7.1 | 13.9 | 44.1 |
> | Ours: Token Filtering | 0.4 | 9.2 | 9.3 | 37.4 | 14.8 | 6.8 | 2.6 | 16.4 | 12.6 | 16.3 | 45.7 |
> | Ours: Normalization | 2.4 | 12.9 | 5.4 | 34.4 | 2.1 | 6.3 | 3.4 | 15.9 | 6.9 | 6.0 | 45.3 |
>
> | Datasets | NarrativeQA | DuReader (zh) | MultiField QA-zh | MultiField QA-en | GovReport | LSHT (zh) | RepoBench-P | Passage Retrieval-en | Passage Retrieval-zh | TREC | avg |
> |-|---|----|---|--|---|----|---|------|----|---|------|
> | Metrics | F1 | Rouge-L | F1 | F1 | Rouge-L | Accuracy | Edit Sim | Accuracy | Accuracy | Accuracy | |
> | Vanilla | 0.3 | 6.3 | 8.9 | 8.2 | 5.2 | 1.0 | 11.4 | 1.2 | 3.3 | 21.0 | 8.1 |
> | DeciMamba | 0.9 | 6.6 | 9.9 | 10.4 | 7.5 | 2.5 | 35.4 | 4.1 | 0.2 | 17.5 | 11.1 |
> | Ours: Token Filtering | 1.8 | 14.7 | 13.4 | 11.2 | 11.2 | 3.5 | 40.3 | 1.7 | 3.7 | 46.5 | 15.0 |
> | Ours: Normalization | 2.3 | 11.1 | 9.5 | 10.3 | 12.4 | 4.9 | 38.5 | 2.6 | 3.8 | 48.0 | 13.5 |
>
> **Tab.B: Perplexity on the PG19 dataset with 1.4B Mamba models.** This table compares the language modeling performance of DeciMamba as well as the two proposed decay adjustment methods.
>
> | Context Length        | 10k   | 20k   | 30k   | 40k   | 50k   | 60k   | 70k   | 80k   | 90k   | 100k  |
> |-----------------------|--------|-------|-------|-------|-------|-------|-------|-------|-------|-------|
> | DeciMamba            | 14.54  | 18.34 | 23.55 | 26.82 | 28.98 | 30.56 | 29.97 | 29.29 | 28.16 | 28.97 |
> | Ours: Token Filtering | 10.46  | 10.75 | 11.05 | 11.04 | 11.53 | 12.24 | 12.35 | 12.24 | 12.11 | 12.17 |
> | Ours: Normalization   | 10.72  | 10.97 | 11.24 | 10.99 | 11.67 | 11.67 | 11.69 | 11.42 | 11.47 | 11.64 |
>
> As observed in Tab. A and Tab. B, the alternative method (“Ours: Normalization”) can consistently improve Mamba’s performance for long-context tasks. For instance, it brings a 5.4% higher average accuracy on LongBench (as shown in the last column of the bottom half of Tab. A) and consistently reduced perplexity compared to DeciMamba on the PG19 dataset (as shown in the second and last row of Tab. B).
>
> This set of experiments further validates the effectiveness and generality of our insight. We believe this insight could shed light on future research on other decay adjustment methods and innovations to enhance SSMs’ long-context capabilities.
>
> ## **Q2: LongMamba shows some improvements for long contexts, but it still seems to be somewhat behind Transformer-based LLMs designed for long contexts. If training were allowed, would the performance be further enhanced?**
>
> Thank you for this great question! We have applied our LongMamba to a 7B Mamba model (i.e., Falcon-Mamba 7B [a]) and compared it with other 7B Transformer [b,c,d] models on LongBench. Specifically, the LongBench results of Transformer models [b,c,d] are the reported ones in the LongBench paper.

---

> ### Author Response · Authors · 2024-11-25
> **Author Response #3**
>
> **Tab. C: Benchmark 7B Mamba and Transformer models on LongBench.** The table compares the performance of the vanilla 7B Falcon-Mamba model [a], LongMamba-enhanced Falcon-Mamba, Llama2-7B-chat-4k [b], XGen-7B-8k[c] and Vicuna-v1.5-7B-16k [d] on the long-context understanding tasks from LongBench. To enhance readability, the table is divided into two sections (top and bottom), with all values expressed as percentages.
>
>
> | Datasets | Passage Count | SAMSum | 2WikiMQA | TriviaQA | Qasper | VCSUM (zh) | Musique | MultiNews | QMSum | HotpotQA | LCC |
> |--------|------|--------|----------|----------|--------|------------|---------|-----------|-------|----------|---------|
> | Metrics | Accuracy | Rouge-L| F1 | F1 | F1 | Rouge-L | F1 | Rouge-L | Rouge-L| F1 | Edit Sim|
> | Vanilla Mamba | 0.1 | 20.0 | 27.6 | 73.1 | 31.0 | 7.2 | 7.4 | 25.8 | 18.5 | 22.6 | 48.4 |
> | LongMamba | 1.5 | 25.1 | 28.2 | 79.9 | 32.1 | 9.9 | 12.1 | 25.8 | 21.5 | 31.8 | 49.5 |
> | Llama2-7B-chat-4k | 2.1 | 40.7 | 32.8 | 77.8 | 19.2 | 0.2 | 9.4 | 25.8 | 20.8 | 25.4 | 52.4 |
> | XGen-7B-8k | 2.1 | 25.3 | 21.1 | 77.8 | 18.1 | 2.2 | 10.3 | 26.2 | 20.5 | 29.7 | 38.6 |
> | Vicuna-v1.5-7B-16k | 6.5 | 40.8 | 20.8 | 86.2 | 26.1 | 15.1 | 9.8 | 27.2 | 22.8 | 25.3 | 51.0 |
>
> | Datasets | NarrativeQA | DuReader (zh) | MultiField QA-zh | MultiField QA-en | GovReport | LSHT (zh) | RepoBench-P | Passage Retrieval-en | Passage Retrieval-zh | TREC | avg |
> |----|-----|-------|-------|------------------|-----------|-----------|-------------|---------------------|---------------------|-------|-------|
> | Metrics | F1 | Rouge-L | F1 | F1 | Rouge-L | Accuracy | Edit Sim | Accuracy | Accuracy | Accuracy | |
> | Vanilla | 5.9 | 12.1 | 20.0 | 30.2 | 22.5 | 2.3 | 38.6 | 4.5 | 3.3 | 70.0 | 23.4 |
> | LongMamba | 13.2 | 11.3 | 20.2 | 30.8 | 23.9 | 12.3 | 42.4 | 4.5 | 3.3 | 70.0 | 26.2 |
> | Llama2-7B-chat-4k | 18.7 | 5.2 | 18.9 | 36.8 | 27.3 | 19.8 | 43.8 | 9.8 | 0.5 | 61.5 | 25.8 |
> | XGen-7B-8k | 18.0 | 11.0 | 14.8 | 37.7 | 27.3 | 20.5 | 38.6 | 8.5 | 3.5 | 65.5 | 24.6 |
> | Vicuna-v1.5-7B-16k | 19.4 | 19.3 | 43.0 | 38.5 | 27.9 | 28.8 | 43.5 | 4.5 | 5.0 | 71.5 | 30.1 |
>
> We can draw the following observations:
> 1. **The vanilla Mamba exhibits lower performance compared to Transformer models** on long context understanding tasks. Specifically, Llama2-7B-chat-4k [b], XGen-7B-8k[c], and Vicuna-v1.5-7B-16k [d] achieve 2.4%, 1.3% and 6.7% higher average accuracy, respectively, compared to the vanilla 7B Falcon-Mamba model (refer to the last column in the bottom half of the table).
> 2. **Our LongMamba can enhance the performance of vanilla Mamba models and narrows the gap with or even surpass Transformer models.** As shown in the last column of the bottom half of the table, LongMamba achieves a 2.8% increase in average accuracy compared to the vanilla Mamba model, outperforming Llama2-7B-chat-4k [b] (+0.4%) and reducing the gap with Vicuna-v1.5-7B-16k [d] from 6.7% to 4.0%.
>
> We believe that this is a positive signal for the research community, implying that Mamba's long-context capability can match, if not surpass, that of Transformers. Our LongMamba approach demonstrates the potential to unlock Mamba's long-context capabilities in a training-free manner, serving as a promising step toward activating these latent abilities and inspiring the development of more sophisticated long-context solutions for Mamba.
>
> In addition, as you suggested, further long-context fine-tuning, which has been shown to be effective for Transformer models [e, f, g], has the potential to further narrow the performance gap. LongMamba is currently designed as a training-free technique for ease of use and can be integrated into the fine-tuning process or applied in a post-fine-tuning manner. Although we don’t have sufficient time to complete this exploration within the rebuttal period, this is a promising direction for future work. We will actively work on this and share our findings with the community once ready.

---

> ### Author Response · Authors · 2024-11-25
> **Author Response #4**
>
> ## **Q3: It would be more convincing if the authors could provide an evaluation of the Ruler benchmark.**
>
> Thank you for the advice! Following your suggestion, we have added the experiments on the RULER benchmark. Specifically, we adopt the Mamba-1.4B model, which was pre-trained on 2k sequence length, and test its performance on both the 8k and 16k sequence lengths provided by the Ruler benchmark. The results are reported in Tab. D and E below, respectively.
>
> We can observe that LongMamba demonstrates notable performance improvements across all kinds of long-context tasks from the Ruler benchmark, highlighting the effectiveness of the proposed framework in addressing diverse long-context understanding challenges.
>
> **Tab. D: Benchmark results on the RULER dataset under a 8192 context length using the 1.4B Mamba model.**
>
> | Tasks                 | niah_single_1 | niah_single_2 | niah_single_3 | niah_multi_key_1 | niah_multivalue | niah_multiquery | vt   | cwe   | fwe   | qa_1  | qa_2  |
> |-----------------------|---------------|---------------|---------------|------------------|-----------------|-----------------|-------|-------|-------|-------|-------|
> | Vanilla Mamba        | 5.0           | 5.0           | 1.0           | 6.0              | 3.5             | 2.5             | 0.2   | 1.6   | 26.7  | 11.0  | 14.0  |
> | LongMamba            | 78.0          | 17.0          | 2.0           | 14.0             | 10.3            | 8.3             | 23.6  | 11.7  | 44.0  | 13.0  | 17.0  |
>
> **Tab. E: Benchmark results on the RULER dataset under a 16384 context length using the 1.4B Mamba model.**
>
> | Tasks                 | niah_single_1 | niah_single_2 | niah_single_3 | niah_multi_key_1 | niah_multivalue | niah_multiquery | vt   | cwe   | fwe   | qa_1  | qa_2  |
> |-----------------------|---------------|---------------|---------------|------------------|-----------------|-----------------|-------|-------|-------|-------|-------|
> | Vanilla Mamba        | 0.0           | 0.0           | 1.0           | 1.0              | 0.5             | 0.0             | 0.0   | 0.0   | 0.0   | 1.0   | 0.0   |
> | LongMamba            | 44.0          | 5.0           | 3.0           | 8.0              | 4.0             | 4.0             | 26.6  | 0.2   | 39.3  | 11.0  | 12.0  |
>
> *For Tab. D and E, we generate 100 sequences for each task. Note that we omit the “niah_multikey_2” and “niah_multikey_3” tasks, because the Mamba-1.4B model achieves 0 accuracy on both tasks even with 2k context length*
>
> -------------
>
> [a] Zuo, J., Velikanov, M., Rhaiem, D. E., Chahed, I., Belkada, Y., Kunsch, G., & Hacid, H. (2024). Falcon mamba: The first competitive attention-free 7b language model. arXiv preprint arXiv:2410.05355.
>
> [b] Touvron, H., Martin, L., Stone, K., Albert, P., Almahairi, A., Babaei, Y., ... & Scialom, T. (2023). Llama 2: Open foundation and fine-tuned chat models. arXiv preprint arXiv:2307.09288.
>
> [c] Nijkamp, E., Xie, T., Hayashi, H., Pang, B., Xia, C., Xing, C., Vig, J., Yavuz, S., Laban, P., Krause, B., Purushwalkam, S., Niu, T., Kryscinski, W., Murakhovs’ka, L., Choubey, P. K., Fabbri, A., Liu, Y., Meng, R., Tu, L., Bhat, M., Wu, C.-S., Savarese, S., Zhou, Y., Joty, S. R., & Xiong, C. (2023). Long sequence modeling with XGen: A 7B LLM trained on 8K input sequence length. arXiv preprint. arXiv:2309.03450
>
> [d] Zheng, L., Chiang, W. L., Sheng, Y., Zhuang, S., Wu, Z., Zhuang, Y., ... & Stoica, I. (2023). Judging llm-as-a-judge with mt-bench and chatbot arena. Advances in Neural Information Processing Systems, 36, 46595-46623.
>
> [e] Chen, Y., Qian, S., Tang, H., Lai, X., Liu, Z., Han, S., & Jia, J. (2024). LongLoRA: Efficient fine-tuning of long-context large language models. arXiv preprint arXiv:2309.12307
>
> [f] Zhang, H. (2024). SinkLoRA: Enhanced efficiency and chat capabilities for long-context large language models. arXiv preprint arXiv:2406.05678
>
> [g] Cai, R., Tian, Y., Wang, Z., & Chen, B. (2024). LoCoCo: Dropping in convolutions for long context compression. arXiv preprint arXiv:2406.05317

---

> > ### Comment · Reviewer_1AHb · 2024-11-27
> > **Thank you for your rebuttal**
> >
> > Thank you for your detailed rebuttal, it addressed most of my concerns. I'll maintain my current positive rating.

---

> > > ### Author Response · Authors · 2024-11-27
> > > **Further response**
> > >
> > > Thank you for recognizing our discovery and root analysis of Mamba's limitations, which we also believe could provide insights for the community and inspire new long-context Mamba solutions. We will follow your suggestion to add the long-context benchmark with transformers and the Ruler results in our final version.

---

### Official Review · Reviewer_P417 · 2024-11-03

**Soundness:** 3
**Presentation:** 4
**Contribution:** 3
**Rating:** 8
**Confidence:** 3

**Summary:**

This paper introduces LongMamba, a training-free techniques to enhance Mamba model's ability to handle long sequences beyond the training context lenght.

The authors discovered that Mamba's hidden state channels can be categorized into two types: local channels that focus on nearby context, and global channels that attend to the entire input sequence.
The key limitation they identified is that global channels fails to be functional when processing sequences longer than the training length and they hypothesize the exponential decay is the root cause.

To this end, they propsoe LongMamba that addresses this by enlarging receptive fields of global channels by filtering out the less important tokens while maintaining similar decay distributions as during training.
Through extensive experiments on tasks like Passkey Retrieval, Document Retrieval, Language Modeling, and LongBench, LongMamba shows promising result without additional training and outperform vanilla mamba and previous methods.

**Strengths:**

- The finding about the categroies of channels of Mamba models(local and global) is interesting and meaningful, which could bring insight to the community for better understanding the Mamba model's behavior.

- The experiment results are strong as it outperforms the vanilla Mamba model by large margin.

- This paper is well-written and easy to follow, for example, they reveal the "categroies of channels" phenomena with sufficient analysis and visualization and figure 2 (the method overview) is tidy and clear

**Weaknesses:**

The paper is generally good but I believe the experiments section can be improved by adding direct comparsion with other long-context models beyond Mamba variants.
I find this is necessary because the core motivation of this work is to address the poor long-context ability of Mambas compared to the transformer-based model, as said in the introduction section. Therefore, it is useful to learn how much LongMamba helps bridge the performance gap. More specifical, the author may follow the [Mamba](https://arxiv.org/abs/2312.00752)'s main experiment setting and compare the LongMamba to varients of transformers (like [Pythia](https://arxiv.org/abs/2304.01373) ) or hypird model

**Questions:**

(1) How does the LongMamba perform on the code categroy of LongBench?

(2) Does the token filtering process bring extra compute or memory cost?

(3) How does the "local and global channels" behave on larger Mamba model?

---

> ### Author Response · Authors · 2024-11-25
> **Author Response #1**
>
> We greatly appreciate your positive feedback and constructive suggestions/questions for our work, and have addressed each of your comments in detail below:
>
> ## **W1: I believe the experiments section can be improved by adding direct comparison with other long-context models beyond Mamba variants.**
>
> Thank you for this great suggestion! We have applied our LongMamba to a 7B Mamba model (i.e., Falcon-Mamba 7B [a]) and compared it with other 7B Transformer [b,c,d] models on LongBench. Specifically, the LongBench results of Transformer models [b,c,d] are the reported ones in the LongBench paper.
>
> **Tab. A: Benchmark 7B Mamba and Transformer models on LongBench:** The table compares the performance of the vanilla 7B Falcon-Mamba model [a], LongMamba-enhanced Falcon-Mamba, Llama2-7B-chat-4k [b], XGen-7B-8k[c] and Vicuna-v1.5-7B-16k [d] on the long-context understanding tasks from LongBench. To enhance readability, the table is divided into two sections (top and bottom), with all values expressed as percentages.
> | Datasets                | Passage Count | SAMSum | 2WikiMQA | TriviaQA | Qasper | VCSUM (zh) | Musique | MultiNews | QMSum | HotpotQA | LCC     |
> |---------|---------------|--------|----------|----------|--------|------------|---------|-----------|-------|----------|---------|
> | Metrics                | Accuracy      | Rouge-L| F1       | F1       | F1     | Rouge-L    | F1      | Rouge-L   | Rouge-L| F1       | Edit Sim|
> | Vanilla Mamba          | 0.1           | 20.0   | 27.6     | 73.1     | 31.0   | 7.2        | 7.4     | 25.8      | 18.5  | 22.6     | 48.4    |
> | LongMamba              | 1.5           | 25.1   | 28.2     | 79.9     | 32.1   | 9.9        | 12.1    | 25.8      | 21.5  | 31.8     | 49.5    |
> | Llama2-7B-chat-4k      | 2.1           | 40.7   | 32.8     | 77.8     | 19.2   | 0.2        | 9.4     | 25.8      | 20.8  | 25.4     | 52.4    |
> | XGen-7B-8k             | 2.1           | 25.3   | 21.1     | 77.8     | 18.1   | 2.2        | 10.3    | 26.2      | 20.5  | 29.7     | 38.6    |
> | Vicuna-v1.5-7B-16k     | 6.5           | 40.8   | 20.8     | 86.2     | 26.1   | 15.1       | 9.8     | 27.2      | 22.8  | 25.3     | 51.0    |
>
> | Datasets                | NarrativeQA | DuReader (zh) | MultiField QA-zh | MultiField QA-en | GovReport | LSHT (zh) | RepoBench-P | Passage Retrieval-en | Passage Retrieval-zh | TREC  | avg   |
> |-------------|-------------|-------|------------------|------------------|-----------|-----------|-------------|---------------------|---------------------|-------|-------|
> | Metrics                | F1          | Rouge-L       | F1               | F1               | Rouge-L   | Accuracy  | Edit Sim    | Accuracy            | Accuracy            | Accuracy |       |
> | Vanilla                | 5.9         | 12.1          | 20.0             | 30.2             | 22.5      | 2.3       | 38.6        | 4.5                 | 3.3                 | 70.0  | 23.4  |
> | LongMamba              | 13.2        | 11.3          | 20.2             | 30.8             | 23.9      | 12.3      | 42.4        | 4.5                 | 3.3                 | 70.0  | 26.2  |
> | Llama2-7B-chat-4k      | 18.7        | 5.2           | 18.9             | 36.8             | 27.3      | 19.8      | 43.8        | 9.8                 | 0.5                 | 61.5  | 25.8  |
> | XGen-7B-8k             | 18.0        | 11.0          | 14.8             | 37.7             | 27.3      | 20.5      | 38.6        | 8.5                 | 3.5                 | 65.5  | 24.6  |
> | Vicuna-v1.5-7B-16k     | 19.4        | 19.3          | 43.0             | 38.5             | 27.9      | 28.8      | 43.5        | 4.5                 | 5.0                 | 71.5  | 30.1  |
>
> We can draw the following observations:
> 1. **The vanilla Mamba exhibits lower performance compared to Transformer models** on long context understanding tasks. Specifically, Llama2-7B-chat-4k [b], XGen-7B-8k[c], and Vicuna-v1.5-7B-16k [d] achieve 2.4%, 1.2% and 6.7% higher average accuracy, respectively, compared to the vanilla 7B Falcon-Mamba model (refer to the last column in the bottom half of the table).
> 2. **Our LongMamba can enhance the performance of vanilla Mamba models and narrows the gap with or even surpass Transformer models**. As shown in the last column of the bottom half of the table, LongMamba achieves a 2.8% increase in average accuracy compared to the vanilla Mamba model, outperforming Llama2-7B-chat-4k (+0.4%) and XGen-7B-8k (+1.6%), and reducing the gap with Vicuna-v1.5-7B-16k from 6.7% to 4.0%.
>
> We believe that this is a positive signal for the research community, implying that Mamba's long-context capability can match, if not surpass, that of Transformers. Our LongMamba approach demonstrates the potential to unlock Mamba's long-context capabilities in a training-free manner, serving as a promising step toward activating these latent abilities and inspiring the development of more sophisticated long-context solutions for Mamba.

---

> ### Author Response · Authors · 2024-11-25
> **Author Response #2**
>
> ## **Q1: How does the LongMamba perform on the code category of LongBench?**
>
> Thank you for the question! We have provided this comparison with both the vanilla Mamba and DeciMamba on the LCC and RepoBench-P tasks (i.e., two coding tasks) in LongBench in Tab. B below. We can observe that LongMamba still achieves higher accuracy than both the vanilla Mamba and DeciMamba, consistently demonstrating its effectiveness. We will add the full benchmark on LongBench in the final version.
>
> **Tab. B: Benchmarking results on the coding tasks from LongBench with the 1.4B Mamba model.** All values are expressed as percentages.
> | Datasets      | LCC            | RepoBench-P     |
> |---------------|----------------|-----------------|
> | Metrics       | Edit Similarity| Edit Similarity |
> | Vanilla Mamba | 40.6           | 11.4            |
> | DeciMamba     | 44.1           | 35.4            |
> | LongMamba     | 45.7           | 40.3            |
>
> ---
>
> ## **Q2: Does the token filtering process bring extra compute or memory cost?**
> Thank you for asking this important question! We clarify that the token filtering process in our method introduces negligible latency and memory overhead. Specifically, following your question, we have tested the latency and memory overhead of the token filtering process in our method using both 130M and 1.4B Mamba models, with a batch size of 1 and context lengths ranging from 2K to 128K tokens. We consistently observe a negligible (<5%) increase in both the prefilling and generation latency, as well as the peak memory usage (reported by the PyTorch memory profiler).
> This minimal impact is due to the highly efficient implementation of the token filtering operation, which selectively sets specific elements in the Δ tensor to zero. Since the Δ tensor is 16 times smaller than the SSM state, operations on it result in minimal additional computational or memory costs compared to the inference process.
>
> We will clarify this in the final version.
>
> ---
>
> ## **Q3: How do the "local and global channels" behave on a larger Mamba model?**
>
> Thank you for the question! To answer your question, we have applied our method to the 7B Falcon-Mamba model [a], which is one of the best-performed Mamba models, adopting the  same 1e-30 threshold for identifying global channels. The results are provided in Tab. A (benchmark on LongBench, in our response to W1 above) and Tab. C (benchmark on PG19 perplexity) below.
>
> **Tab. C: Perplexity on PG19 with different context lengths with 7B Falcon-Mamba [a].**
>
> | Context Length | 10k  | 20k   | 30k   | 40k   | 50k    | 60k     | 70k      |
> |----------------|-------|-------|-------|-------|--------|---------|----------|
> | Vanilla Mamba  | 9.11  | 13.52 | 34.27 | 25.64 | 145.31 | 611.85  | 1574.17  |
> | LongMamba      | 8.99  | 9.34  | 10.16 | 9.47  | 9.67   | 10.82   | 11.17    |
>
> The results in Tab. A and Tab. C consistently demonstrate that at the 7B model scale, LongMamba can still notably enhance the performance of Mamba models on long-context tasks. For example, LongMamba achieves a 2.8% increase in average accuracy compared to the vanilla 7B Mamba model on LongBench in Tab. A and enables consistently lower perplexity on PG19, especially under longer sequences, in Tab. C.
>
> -----------------------------------------
>
> [a] Zuo, J., Velikanov, M., Rhaiem, D. E., Chahed, I., Belkada, Y., Kunsch, G., & Hacid, H. (2024). Falcon mamba: The first competitive attention-free 7b language model. arXiv preprint arXiv:2410.05355.
>
> [b] Touvron, H., Martin, L., Stone, K., Albert, P., Almahairi, A., Babaei, Y., ... & Scialom, T. (2023). Llama 2: Open foundation and fine-tuned chat models. arXiv preprint arXiv:2307.09288.
>
> [c] Nijkamp, E., Xie, T., Hayashi, H., Pang, B., Xia, C., Xing, C., Vig, J., Yavuz, S., Laban, P., Krause, B., Purushwalkam, S., Niu, T., Kryscinski, W., Murakhovs’ka, L., Choubey, P. K., Fabbri, A., Liu, Y., Meng, R., Tu, L., Bhat, M., Wu, C.-S., Savarese, S., Zhou, Y., Joty, S. R., & Xiong, C. (2023). Long sequence modeling with XGen: A 7B LLM trained on 8K input sequence length. arXiv preprint. arXiv:2309.03450
>
> [d] Zheng, L., Chiang, W. L., Sheng, Y., Zhuang, S., Wu, Z., Zhuang, Y., ... & Stoica, I. (2023). Judging llm-as-a-judge with mt-bench and chatbot arena. Advances in Neural Information Processing Systems, 36, 46595-46623.

---

> > ### Comment · Reviewer_P417 · 2024-11-26
> >
> > Thank you for your informative response, which addressed all of my concerns. I decided to keep my positive score.

---

> > > ### Author Response · Authors · 2024-11-26
> > > **Further response**
> > >
> > > Thank you for recognizing the interesting and meaningful aspects of our work and the insights it could provide to the community! Following your insightful suggestion, we will include a comparison with transformer models in our final version, as well as add results for larger Mamba models and all LongBench tasks.

---

### Official Review · Reviewer_xMy1 · 2024-11-04

**Soundness:** 3
**Presentation:** 4
**Contribution:** 2
**Rating:** 5
**Confidence:** 5

**Summary:**

The paper introduces LongMamba, a method to extend the context length of Mamba models without additional training, focusing on long-context understanding for state-space models. Mamba models, though efficient in handling long contexts due to their linear complexity, struggle with context lengths far beyond training sequences due to exponential decay in the attention value, especially toward early tokens. LongMamba addresses this by categorizing channels into local and global based on receptive field lengths, then enlarging global channels' receptive fields via adaptive decay adjustments through token filtering. This modification allows LongMamba to maintain performance over long contexts without the computational cost of fine-tuning, making it a practical improvement over previous methods like DeciMamba.

**Strengths:**

1 - The paper addresses a critical challenge in long-context modeling, particularly for SSMs like Mamba, which suffer from limited long-sequence capabilities despite their efficient state-space mechanics. Enhancing this capability without retraining aligns well with real-world demands for scalable, efficient language models.


2 - Devising a “training-free” context extension technique is an important aspect of the paper, as directly training the SSMs on longer context requires significant compute/resource and expansive datasets.


3 - Extensive benchmarks show that LongMamba consistently outperforms vanilla Mamba on long-context tasks, including language modeling on the PG-19 dataset and passkey/document retrieval. LongMamba’s results in real-world tasks, such as LongBench, also showcase its adaptability

**Weaknesses:**

1- A substantial limitation is that LongMamba relies heavily on the DeciMamba[1] framework, raising questions about its independent novelty. Although LongMamba innovatively extends global channels, the reliance on DeciMamba's token handling might limit its perceived originality. In particular, the idea of token filtering based on the averaged delta_t value was proposed by DeciMamba. LongMamba merely improves upon this idea by limiting the scope of this averaging to some specific channels (global channels).

2 -The paper does not systematically address threshold selection for distinguishing global versus local channels. This selection is left to empirical values, which may lack generalizability across different datasets or models. A more formal approach to tuning this threshold would strengthen LongMamba's applicability. This is of importance, as global and local channels are integral parts of the LongMamba framework.


3 - The method for handling global channels’ decay distribution adjustment is not fully elaborated. It remains ambiguous whether averaging across global channels fully mitigates decay issues over extended contexts, potentially affecting LongMamba’s consistency in varied contexts (Figure 2c, Decay Distribution Adjustment).

**Questions:**

1. Could the authors please address the points mentioned in the weaknesses section?

2. Is a similar exponential decay observed in the Mamba2[2] model? If so, does LongMamba improve the long-context understanding in this architecture as well?

3. Would the proposed approach generalize to other SSMs beyond Mamba, including hybrid Transformer/SSM architectures, or is it specifically tailored to Mamba’s architectural nuances?

4. LongBench is a comprehensive benchmark. How does LongMamba perform on other LongBench tasks, namely 2wikimqa, TREC, TriviaQA, LCC, and RepoBench-P?

5. Have the authors considered alternative strategies for decay adjustment besides token filtering? If so, why was filtering chosen over other methods?

References:

[1]Ben-Kish, Assaf, et al. "DeciMamba: Exploring the Length Extrapolation Potential of Mamba." arXiv preprint arXiv:2406.14528 (2024).

[2]Dao, Tri, and Albert Gu. "Transformers are SSMs: Generalized models and efficient algorithms through structured state space duality." arXiv preprint arXiv:2405.21060 (2024).

---

> ### Author Response · Authors · 2024-11-25
> **Author Response #1**
>
> We greatly appreciate the insightful comments and the concerns you raised, and have addressed them below:
>
> ---
>
> ## **W1 & Q5: LongMamba relies heavily on the DeciMamba framework (specifically, the token filtering idea), raising questions about its independent novelty. Have the authors considered alternative strategies for decay adjustment?**
>
> We humbly clarify that the key contribution of LongMamba lies in the insight distinguishing **global** and **local** channels in Mamba, which serves as the foundation for enhancing its long-context capabilities. While we use token filtering to implement this insight, we emphasize that it is just one of many possible approaches to leverage or implement our insight, showcasing the broader applicability and originality of our contribution. Specifically, our new contributions beyond DeciMamba are as follows:
>
> **1. Key Insight:**
>
> We discover that the exponential hidden state decay of the global channels is the primary cause of Mamba’s constrained receptive field and limited performance on long-context tasks. This finding is also highly recognized by other reviewers: Reviewer P417 remarked, "The finding about the categroies of channels of Mamba models (local and global) is interesting and meaningful, which could bring insight to the community for better understanding the Mamba model's behavior", Reviewer 1AHb highlighted, “the discovery and explanation of Mamba models’ limited performance in long contexts are quite interesting," and Reviewer GwXu stated, “the main result is very interesting.”
>
> **2. Token Filtering Implementation:**
>
> Leveraging this key insight, we propose specially handling the global channels to enlarge their receptive field by applying selective token filtering exclusively to the global channels while preserving the locality of the local channels. This strategy has outperformed DeciMamba and maintained the same language modeling perplexity on contexts 10x longer, further validating the **significance and practicality** of our key insight. This has also been recognized by other reviews, such as “Overall a simple mechanism that seemingly enables Mamba to generalize to much longer contexts” by Reviewer GwXu.
>
> **3. Alternative Implementation:**
>
> To further demonstrate the generality of our insight, during the rebuttal period, we implemented and validated an alternative method.
>
> This method dubbed “decay normalization” adjusts the decay by normalizing $\Delta_t$ for all tokens in the same global channel with a scalar constant c (i.e., $\Delta’_t = \Delta_t / c$), and subsequently derives the adjusted $\bar{A}’_t$ and $\bar{B}’_t$ using Eq.7 of the submitted manuscript. The factor $c=c(S)$ is computed to satisfy Eq.16 in Section 5.2 of the submitted manuscript, similar to how we obtain $g(S)$.
>
> We evaluated this alternative decay adjustment method on both the LongBench and PG19 tasks. The results are summarized in Tab. A and Tab. B below, where “Ours: Token Filtering” corresponds to the decay adjustment method described in the submitted manuscript, and “Ours: Normalization” represents the alternative approach proposed above.

---

> ### Author Response · Authors · 2024-11-25
> **Author Response #2**
>
> **Tab. A: Benchmarking results on LongBench dataset with 1.4B Mamba models.** This table compares the LongBench performance achieved by the vanilla Mamba model, DeciMamba, as well as the two proposed methods that specially handle the global channels stemming from our insight. To enhance readability, the table is divided into two sections (top and bottom), with all values expressed as percentages.
>
> | Datasets | Passage Count | SAMSum | 2WikiMQA | TriviaQA | Qasper | VCSUM (zh) | Musique | MultiNews | QMSum | HotpotQA | LCC |
> |----|--|--|----|----|---|----|---|---|---|---|---|
> | Metrics | Accuracy | Rouge-L | F1 | F1 | F1 | Rouge-L | F1 | Rouge-L | Rouge-L| F1 | Edit Sim|
> | Vanilla Mamba | 0.4 | 3.7 | 8.6 | 11.4 | 13.6 | 2.1 | 0.4 | 15.7 | 1.1 | 4.7 | 40.6 |
> | DeciMamba | 0.1 | 8.7 | 6.5 | 23.4 | 14.2 | 3.2 | 1.4 | 15.8 | 7.1 | 13.9 | 44.1 |
> | Ours: Token Filtering | 0.4 | 9.2 | 9.3 | 37.4 | 14.8 | 6.8 | 2.6 | 16.4 | 12.6 | 16.3 | 45.7 |
> | Ours: Normalization | 2.4 | 12.9 | 5.4 | 34.4 | 2.1 | 6.3 | 3.4 | 15.9 | 6.9 | 6.0 | 45.3 |
>
> | Datasets | NarrativeQA | DuReader (zh) | MultiField QA-zh | MultiField QA-en | GovReport | LSHT (zh) | RepoBench-P | Passage Retrieval-en | Passage Retrieval-zh | TREC | avg |
> |-|---|----|---|--|---|----|---|------|----|---|------|
> | Metrics | F1 | Rouge-L | F1 | F1 | Rouge-L | Accuracy | Edit Sim | Accuracy | Accuracy | Accuracy | |
> | Vanilla | 0.3 | 6.3 | 8.9 | 8.2 | 5.2 | 1.0 | 11.4 | 1.2 | 3.3 | 21.0 | 8.1 |
> | DeciMamba | 0.9 | 6.6 | 9.9 | 10.4 | 7.5 | 2.5 | 35.4 | 4.1 | 0.2 | 17.5 | 11.1 |
> | Ours: Token Filtering | 1.8 | 14.7 | 13.4 | 11.2 | 11.2 | 3.5 | 40.3 | 1.7 | 3.7 | 46.5 | 15.0 |
> | Ours: Normalization | 2.3 | 11.1 | 9.5 | 10.3 | 12.4 | 4.9 | 38.5 | 2.6 | 3.8 | 48.0 | 13.5 |
>
> **Tab. B: Perplexity on the PG19 dataset with 1.4B Mamba models.** This table compares the language modeling performance of DeciMamba as well as the two proposed decay adjustment methods.
>
> | Context Length        | 10k   | 20k   | 30k   | 40k   | 50k   | 60k   | 70k   | 80k   | 90k   | 100k  |
> |-----------------------|--------|-------|-------|-------|-------|-------|-------|-------|-------|-------|
> | DeciMamba            | 14.54  | 18.34 | 23.55 | 26.82 | 28.98 | 30.56 | 29.97 | 29.29 | 28.16 | 28.97 |
> | Ours: Token Filtering | 10.46  | 10.75 | 11.05 | 11.04 | 11.53 | 12.24 | 12.35 | 12.24 | 12.11 | 12.17 |
> | Ours: Normalization   | 10.72  | 10.97 | 11.24 | 10.99 | 11.67 | 11.67 | 11.69 | 11.42 | 11.47 | 11.64 |
>
> As observed in Tab. A and Tab. B, the alternative method (“Ours: Normalization”) can consistently improve Mamba’s performance for long-context tasks. For instance, it brings a 5.4% higher average accuracy on LongBench (as shown in the last column of the bottom half of Tab. A) and consistently reduced perplexity compared to DeciMamba on the PG19 dataset (as shown in the second and last row of Tab. B).
>
> This set of experiments further validates the effectiveness and generality of our insight. We believe this insight could shed light on future research on other decay adjustment methods and innovations to enhance SSMs’ long-context capabilities.
>
> ---
>
> ## **W2: The paper does not systematically address threshold selection for distinguishing global versus local channels. This selection is left to empirical values, which may lack generalizability across different datasets or models.**
>
> We would like to humbly clarify that the threshold is determined using a systematic strategy for each model, and is then fixed and generalizable across all evaluation tasks for this particular model. In other words, for all experiments throughout the submitted manuscript, we use the same selection threshold (1e-30).
>
> The rationale behind this approach is that we empirically find the optimal threshold generalizes well across tasks. We conjecture that this is because the distinction between global and local channels is not sensitive, as the top global channels might dominate the contributions due to the exponential decay effects. This allows us to rapidly determine the optimal threshold on one task and reuse it across all evaluation tasks for a given model. To demonstrate this, we calibrate the threshold using the Passkey Retrieval task through a grid search (with {1e-20, 1e-25, 1e-30, 1e-35, 1e-40} as the candidates) and reuse the identified optimal value for all other tasks. As shown in Fig. 4 and Tab. 1 of the submitted manuscript, the resulting performance of our method already outperforms vanilla Mamba and DeciMamba, demonstrating the effectiveness of this strategy. Additionally, the results achieved on other tasks, such as those on LongBench and PG19 shown in Tab. A and Tab. B in our response above, also adopt this simple strategy and demonstrate consistent improvements.
>
> Following your suggestion, we have added a clarification for the threshold calibration process in Appendix A of the revised manuscript.

---

> ### Author Response · Authors · 2024-11-25
> **Author Response #3**
>
> ## **W3: The method for handling global channels’ decay distribution adjustment is not fully elaborated. It remains ambiguous whether averaging across global channels fully mitigates decay issues over extended contexts, potentially affecting LongMamba’s consistency in varied contexts (Figure 2c, Decay Distribution Adjustment).**
>
> First, we humbly clarify that our method does not perform cross-channel averaging across all the global channels. Instead, we apply token filtering independently for each channel, ensuring that the exponential decay is effectively addressed for all global channels. Specifically, as introduced in Section 5.2 of our submitted manuscript, we first obtain the distribution of $\bar{A_t}$ for each channel by profiling a subset of the training sequences and compute a threshold $g(S)$ for each global channel, which is used for their decay alignment by filtering out tokens where $\Delta_t < g(S)$ before the hidden state update.
>
> In addition, to demonstrate the effectiveness of our algorithm in mitigating the exponential decay issue, we present the per-channel cumulative decay (as defined in Eq. 14) for Mamba models, with and without LongMamba, in Appendix B.2 of the revised manuscript. As illustrated in these figures, LongMamba can significantly reduce the exponential cumulative decay in the global channels and thus extend the effective receptive field. Detailed discussion on this observation is provided in Appendix B.2 of our revised manuscript.
>
> ---
>
> ## **Q2: Is a similar exponential decay observed in the Mamba2? Does LongMamba improve the long context understanding with Mamba2?**
>
> Thank you for the insightful question! Yes, we have also observed that Mamba2 exhibits a similar exponential decay issue and illustrated the cumulative decay of the Mamba2 model in Appendix B.3 of the revised manuscript.
>
> Following your suggestion, we have also applied LongMamba to Mamba2 and find that LongMamba can notably improve the long-context understanding of Mamba2, with experimental results presented in Tab. C and Tab. D below for the PG19 and LongBench datasets, respectively.
>
> **Tab. C: Perplexity on the PG19 dataset with 1.3B Mamba2 models pre-trained on 2k sequence length.** This table compares the performance of vanilla Mamba2 model and LongMamba under different test sequence lengths.
>
> | Context Length | 10K | 20K | 30K | 40K | 50K | 60K | 70K | 80K | 90K | 100K |
> |------|-----|---------|---------|---------|---------|---------|---------|---------|---------|---------|
> | Vanilla Mamba | 227.65 | 1980.04 | 4213.87 | 3851.41 | 2343.61 | 1531.66 | 1119.94 | 969.64 | 835.21 | 672.14 |
> | LongMamba | 13.71 | 14.8 | 15.4 | 16.07 | 15.19 | 16.15 | 17.28 | 16.62 | 16.28 | 17.88 |
>
> **Tab.D: Benchmark results on LongBench with the 1.3B Mamba2 model pre-trained on 2k sequence length.** This table compares the performance of vanilla Mamba models and LongMamba, under different test sequence lengths. To enhance readability, the table is divided into two sections (top and bottom), with all values expressed as percentages.
>
> | Datasets | Passage Count | SAMSum | 2WikiMQA | TriviaQA | Qasper | VCSUM (zh) | Musique | MultiNews | QMSum | HotpotQA | LCC |
> |----------------|---------------|--------|----------|----------|--------|------------|---------|-----------|-------|----------|---------|
> | Metrics | Accuracy | Rouge-L| F1 | F1 | F1 | Rouge-L | F1 | Rouge-L | Rouge-L | F1 | Edit Sim|
> | Vanilla Mamba | 0.4 | 4.9 | 3.1 | 10.0 | 3.1 | 0.5 | 0.6 | 20.6 | 6.7 | 2.1 | 44.5 |
> | LongMamba | 0.8 | 26.5 | 9.2 | 39.7 | 6.0 | 6.5 | 1.9 | 21.6 | 16.0 | 5.1 | 49.7 |
>
> | Datasets | NarrativeQA | DuReader (zh) | MultiFieldQA-zh | MultiFieldQA-en | GovReport | LSHT (zh) | RepoBench-P | PassageRetrieval-en | PassageRetrieval-zh | TREC | avg |
> |----------------|-------------|---------------|------------------|------------------|-----------|-----------|-------------|---------------------|---------------------|-------|-------|
> | Metrics | F1 | Rouge-L | F1 | F1 | Rouge-L | Accuracy | Edit Sim | Accuracy | Accuracy | Accuracy | |
> | Vanilla | 0.9 | 1.4 | 3.9 | 5.8 | 5.5 | 0.0 | 26.9 | 0.0 | 0.1 | 17.0 | 7.5 |
> | LongMamba | 2.5 | 14.3 | 12.7 | 13.1 | 12.0 | 8.5 | 45.6 | 2.2 | 4.4 | 32.5 | 15.7 |
>
> Result summary: We can observe that LongMamba also enhances the performance of the Mamba2-1.3B model on both datasets. Specifically, as shown in Tab. C, LongMamba can consistently and significantly reduce perplexity across all evaluated context lengths (which range from 10k to 100k tokens). Meanwhile, Tab. D demonstrates that LongMamba improves the average accuracy on LongBench by 8.2% (refer to the last column of the bottom half of Tab. D).

---

> ### Author Response · Authors · 2024-11-25
> **Author Response #4**
>
> ## **Q4: How does LongMamba perform on other LongBench tasks, namely 2WikiMQA, TREC, TriviaQA, LCC, and RepoBench-P?**
>
> Thank you for raising this question! The short answer is that LongMamba consistently performs well in these tasks as well. Specifically, we have provided the benchmark results for all LongBench tasks in Tab. A and Tab. D (in our response to you above) for the 1.4B Mamba model and the 1.3B Mamba2 model, respectively. In addition, following your suggestion, the performance on {2WikiMQA, TREC, TriviaQA, LCC, and RepoBench-P} with the 1.4B Mamba model is listed below in Tab. E:
>
> **Tab. E: Benchmarking results on LongBench dataset with the 1.4B Mamba model.** Note that this table is a subset of Tab. A.
>
> |  Datasets      | 2WikiMQA | TREC  | TriviaQA | LCC   | RepoBench-P |
> |----------------|----------|-------|----------|-------|-------------|
> | Metrics        | F1       | Accuracy | F1       | Edit Sim | Edit Sim    |
> | Vanilla Mamba  | 8.6      | 21.0  | 11.4     | 40.6  | 11.4        |
> | LongMamba      | 9.3      | 46.5  | 37.4     | 45.7  | 40.3        |
>
> As shown in Tab. E, we can observe that the proposed LongMamba can notably improve the performance of Mamba models across all tasks (the accuracy in the last row is consistently higher than that in the third row), again demonstrating the effectiveness of the proposed method.
>
> ---
>
> We are still working on addressing **Q3: Would the proposed approach generalize to other SSMs beyond Mamba, including hybrid Transformer/SSM architectures, or is it specifically tailored to Mamba’s architectural nuances?** We will post a response to this question before the end of the rebuttal period. (*Update: This question has been addressed in “Author Response #5” below.*)

---

> > ### Author Response · Authors · 2024-11-30
> > **Author Response #5**
> >
> > Thank you for the patience, we have addressed Q3 below:
> >
> > ## **Q3: Would the proposed approach generalize to other SSMs beyond Mamba, including hybrid Transformer/SSM architectures, or is it specifically tailored to Mamba’s architectural nuances?**
> >
> > Thank you for this insightful question! Following your suggestion, we have applied LongMamba to the recently released hybrid Transformer/SSM model Hymba [a], which is among the best hybrid models according to its reported results, and benchmark on LongBench-e.
> >
> > The results are detailed in Tab. F, and we observe the following:
> >
> > 1. **LongMamba can enhance long-context understanding capabilities** on 11 out of 13 tasks and maintain comparable performance on the remaining 2 tasks, compared to vanilla Hymba. For instance, on the LCC task, LongMamba achieves a notable score improvement of 7.65% over vanilla Hymba.
> >
> > 2. **The improvement on the hybrid model is moderate**, with an average performance increase of 1.26%. As analyzed in Hymba’s Fig. 7 and Sec. 2.4, we hypothesize that this is because, in hybrid models, the Transformer component predominantly captures global context, while the SSM component focuses more on local context, making the long-context enhancement on SSMs less effective.
> >
> > In summary, these findings suggest that LongMamba remains effective across hybrid architectures, with the degree of impact varying based on the extent to which the SSM components contribute to long-context modeling.
> >
> > **Tab. F: Evaluation results of applying LongMamba to Hymba on LongBench-E.** To enhance readability, the table is divided into two sections (top and bottom), with all values expressed as percentages.
> >
> > | Datasets | Qasper | MultiFieldQA-en | HotpotQA | 2WikiMQA | TREC | TriviaQA | SAMSum |
> > |------------|--------|-----------------|----------|----------|-------|----------|---------|
> > | Metric | F1 | F1 | F1 | F1 | Accuracy | F1 | Rouge-L |
> > | Vanilla | 7.27 | 16.71 | 7.39 | 7.26 | 52.33 | 75.94 | 33.74 |
> > | LongMamba | 7.86 | 17.65 | 7.96 | 8.18 | 53.75 | 76.14 | 35.08 |
> >
> > | Datasets | Passage Count | PassageRetrieval-en | LCC | RepoBench-P | GovReport | MultiNews | Average |
> > |------------|---------------|---------------------|---------|-------------|-----------|-----------|---------|
> > | Metric | Accuracy | Accuracy | Edit Sim| Edit Sim | Rouge-L | Rouge-L | |
> > | Vanilla | 3.08 | 5.93 | 29.87 | 38.87 | 17.62 | 16.01 | 24.00 |
> > | LongMamba | 3.00 | 5.99 | 37.52 | 40.80 | 17.56 | 16.85 | 25.26 |
> >
> > [a] Dong, X., Fu, Y., Diao, S., Byeon, W., Chen, Z., Mahabaleshwarkar, A. S., ... & Molchanov, P. (2024). Hymba: A Hybrid-head Architecture for Small Language Models. arXiv preprint arXiv:2411.13676.

---

### Official Review · Reviewer_GwXu · 2024-11-05

**Soundness:** 3
**Presentation:** 3
**Contribution:** 3
**Rating:** 8
**Confidence:** 3

**Summary:**

The paper proposes a simple modification to Mamba architecture where they prevent channels from exponential decaying by filtering out tokens from the training sequence if the update is small than threshold , for smaller updates and keepin them as is for larger updates. This allows to prevent catastrophic forgetting, and surprisingly allows to reuse existing model on a significantly longer context without finetuning.
The modification seem pretty straightforward where they essentially identify "Global channels" that seem to be particularly prone to exponential decay and change the SSM update procedure to ignore some of the smaller update to reduce the amount of exponential decay for longer contexts.

**Strengths:**

The main result is very interesting: they show that existing SSM architectures acan be expanded to much longer contexts without having to fine tune it. Overall  a simple mechanism that seemingly enables Mamba to generalize to much longer contexts. Great introduction to Mamba models.

**Weaknesses:**

See questions. I think the empirical analysis section would benefit from some more detail study  of distribution of local vs global channels receptive field (see Q5 below). Sec 5 would benefit from expanding how A' are computed.

**Questions:**

1. End of Section 5: line 333, i think you meant Eq 16 not E 18?
2. It is not clear what A'_t are if updates is greater than threshold.  (e.g. dual of Eq 17)
3. How is g(S) computed? Is it computed to satisfy equation 16?
4. Local/Global channel: is the number 0-300-600-900-1200 are just percentiles of a fixed model? It would be better to specifically say that.
5. Can you include a graph showing the cumulative graph of receptive fields for each each channel? Especially this would be interesting for training seq.len and a much longer seq len?

---

> ### Author Response · Authors · 2024-11-25
> **Author Response #1**
>
> We greatly appreciate your encouraging recognition of our contribution and your constructive questions and suggestions, and have addressed the questions below:
>
> ------
>
> ### **Q1: End of Section 5: line 333, I think you meant Eq. 16, not Eq. 18?**
>
> Thank you for catching this typo! You are correct, and we have updated the line in the revised manuscript to use “Eq. 16”.
>
> -----
>
> ### **Q2: What $\bar{A}'_t$ is if updates are greater than the threshold?**
>
> If $\bar{A}’_t$ is greater than the threshold, their values remain unchanged  (i.e., the same as $\bar{A_t}$). We have clarified this in line 332 of the revised manuscript.
>
> ------
>
> ### **Q3: How is $g(S)$ computed? Is it computed to satisfy Eq. 16?**
>
> Yes, you are correct. $g(S)$ is computed to satisfy Eq. 16, aiming to ensure that the cumulative decay on the identified global channels at the test sequence length $S$ aligns with that at the training sequence length $L$ when $S>L$.
>
> To further clarify this, we have reorganized our proposed solution and its corresponding implementation, summarizing them as follows. We will further clarify this in the final version.
>
> **1. Our proposed two-step solution, as detailed in Section 5.2 of our submitted manuscript:**
>
> Step-1: We first  obtain the distribution of $\bar{A_t}$ by profiling on a subset of training sequences.
>
> Step-2: For the target test sequence length $S > L$, we compute a threshold $g(S)$ that satisfies Eq.16 in our submitted manuscript.
>
> **2. The detailed implementation:**
>
> As outlined in Sec. 6.1 of the submitted manuscript, we sample 10 training sequences to obtain $\bar{A_t}$ distribution in Step-1 and build a lookup table by computing $g(S)$ for every 1,000-token increase in context length (i.e., S={1k, 2k, 3k, …, nk, …}) in Step-2. During inference, the test sequence length $S$ is rounded up to the nearest entry in the lookup table, and the corresponding threshold $g(S)$ is retrieved from the lookup table.
>
> ------
>
> ### **Q4: Local/Global channel: Are the numbers 0-300-600-900-1200 just percentiles of a fixed model?**
>
> Thank you for asking this. We would like to clarify that the numbers (0-300-600-900-1200) in Fig. 1 of our submitted manuscript represent the channel indices in the model's weight tensor, not any rankings or percentiles. We selected three specific local channels and two global channels as examples to illustrate these concepts in Fig. 1. This has been clarified in the revised manuscript.
>
> ------
>
> ### **Q5: Can you include a graph showing the cumulative graph of receptive fields for each channel? Especially this would be interesting for training sequence length and a much longer sequence length.**
>
> Thank you for the suggestion! We have added the suggested figure in the revised Appendix B.1. From the figure, it can be observed that for global channels: (1) when the sequence length is smaller than the training sequence length, the receptive field grows linearly with increasing sequence length, and (2) beyond the training sequence length, the receptive field no longer increases linearly and sometimes even decreases, limiting the model’s ability to capture global information from longer sequences.

---

### Author Response · Authors · 2024-12-02

Dear ACs and Reviewers,

We sincerely appreciate the time and effort of all reviewers in providing thoughtful and constructive feedback, especially considering the high volume of submissions at a top-tier conference like ICLR. We also hope that you and your families have had a joyful Thanksgiving holiday.

We are encouraged by the reviewers’ recognition of LongMamba for its key insight, extensive experiments, and clear writing. In particular, we are pleased that our insight—highlighting the exponential hidden state decay of global channels as the primary factor constraining Mamba’s receptive field and performance on long-context tasks—has been well received.

In addition to the commendations, we also received insightful questions from reviewers and have addressed all of them in our response to each reviewer. We welcome and look forward to further discussions with all reviewers, especially with tomorrow being the rebuttal deadline.

In the sections below, we provide a brief summary of our responses, particularly highlighting two common insightful questions from reviewers: the benchmark of LongMamba-enhanced Mamba with Transformers and the novelty of LongMamba.

---

## **Benchmark LongMamba-enhanced Mamba with Transformer Models:**

Following the suggestion of Reviewer P417, we benchmarked a 7B Mamba model against Transformer models of similar scale during the rebuttal period. **While the vanilla Mamba model showed lower performance compared to Transformer models, LongMamba demonstrated a significant improvement, narrowing the performance gap** and even surpassing certain Transformer models. Detailed experimental results are included in our response to Reviewer P417’s W1.

---

## **Novelty of LongMamba:**

**The core contribution of LongMamba lies in its insight** that the exponential hidden state decay of global channels is the primary cause of Mamba’s constrained receptive field and limited performance on long-context tasks. Building on this insight, LongMamba specially handles the identified global channels with selective token filtering, which has demonstrated superior performance compared to the prior work DeciMamba [a].

Moreover, **we have validated the generality and applicability** of our insight through the following experiments conducted during the rebuttal period:

1. We have implemented an alternative method to instantiate our insight (i.e., “decay normalization”, as discussed in our response to Reviewer xMy1’s W1 & Q5), which also enhances Mamba’s long-context understanding capabilities.
2. Applying LongMamba to the Mamba-2 [b] model and the hybrid Transformer/SSM model Hymba [c] also improves their long-context understanding capabilities. Detailed experimental results are provided in our responses to Reviewer xMy1’s Q2 and Q3, respectively.

---

> ### Author Response · Authors · 2024-12-02
>
> ## **Summary of All Responses:**
>
> ### **Added Visualizations:**
>
> 1. Receptive fields under different context lengths
>    - Added in Appendix B.1 of the revised manuscript.
>
> 2. Effect of LongMamba on mitigating the exponential hidden state decay
>    - Included in Appendix B.2 of the revised manuscript.
>
> 3. Hidden state decay factor in Mamba-2
>    - Provided in Appendix B.3 of the revised manuscript.
>
> ### **Added Experiments:**
>
> 1. Proposing and validating an alternative method to implement our key insight
>    - Results in the response to Reviewer xMy1’s W1 & Q5.
>
> 2. Extending LongMamba to Mamba-2
>    - Results in the response to Reviewer xMy1’s Q2.
>
> 3. Performance on additional LongBench tasks
>    - Full benchmark results in Tab. A of the author response #2 to Reviewer xMy1.
>
> 4. Applying LongMamba to hybrid Transformer/SSM architectures
>    - Results in the response to Reviewer xMy1’s Q3.
>
> 5. Comparison with Transformer models
>    - Results in the response to Reviewer P417’s W1.
>
> 6. Measuring the compute and memory overhead of LongMamba
>    - Results in the response to Reviewer P417’s Q3.
>
> 7. Extending LongMamba to larger Mamba models
>    - Results in the response to Reviewer P417’s Q3.
>
> 8. Benchmarking on the RULER dataset
>    - Results in the response to Reviewer 1AHb’s Q3.
>
> ### **Clarifications and Fixed Writings:**
>
> 1. Discussion of our work’s novelty and differences from previous works
>    - Included in the response to Reviewer xMy1’s W1 & Q5.
>
> 2. Systematic threshold selection process
>    - Detailed in Appendix A of the revised manuscript.
>
> 3. Algorithm for decay adjustment
>    - Explained in the response to Reviewer xMy1’s W3.
>
> 4. Miscited equation in Sec. 5, Line 333
>    - Corrected in the revised manuscript.
>
> 5. Clarification on \bar{A’_t} updates
>    - Provided in Sec. 5.2 of the revised manuscript.
>
> 6. Derivation of g(S)
>    - Algorithm detailed in the response to Reviewer GwXu’s Q3.
>
> 7. Channel index definition in Fig. 1
>    - Updated in the caption of Fig. 1 in the revised manuscript.
>
> ---
>
> [a] Ben-Kish, Assaf, et al. "DeciMamba: Exploring the Length Extrapolation Potential of Mamba." arXiv preprint arXiv:2406.14528 (2024)
>
> [b] Dao, Tri, et al. “Transformers are SSMs: Generalized Models and Efficient Algorithms Through Structured State Space Duality.” arXiv preprint arXiv:2405.21060 (2024)
>
> [c] Dong, Xin, et al. “Hymba: A Hybrid-head Architecture for Small Language Models.” arXiv preprint arXiv:2411.13676. (2024)

---

### Meta-Review · Area_Chair_XF4P · 2024-12-23

**Metareview:**

All reviewers found the work interesting, with strong results. The proposed technique for extending Mamba’s receptive field is both novel and clearly explained, supported by thorough experiments on multiple tasks. In particular, mitigating exponential decay by filtering out smaller updates demonstrates a straightforward yet impactful solution to catastrophic forgetting, without finetuning. Their experiments highlight the effectiveness of this approach in real-world settings. The paper is well-written, and its insights on local and global channels are valuable. Overall, a clear accept.

**Additional Comments On Reviewer Discussion:**

n/a

---

### Decision · Program_Chairs · 2025-01-22

Accept (Poster)